# ERK3/MAPK6 controls IL-8 production and chemotaxis

**Katarzyna Bogucka[1], Malvika Pompaiah[1], Federico Marini[2,3], Harald Binder[2,4], Gregory Harms[1,5], Manuel Kaulich[6,7], Matthias Klein[8], Christian Michel[9], Markus P Radsak[9], Sebastian Rosigkeit[1], Peter Grimminger[10], Hansjörg Schild[8], Krishnaraj Rajalingam[1,11]\***

[1]Cell Biology Unit, University Medical Center of the Johannes Gutenberg University Mainz, Mainz, Germany; [2]Institute of Medical Biostatistics, Epidemiology and Informatics, University Medical Center of the Johannes Gutenberg University Mainz, Mainz, Germany; [3]Center for Thrombosis and Hemostasis (CTH), University Medical Center of the Johannes Gutenberg University Mainz, Mainz, Germany; [4]Institute of Medical Biometry and Statistics, Faculty of Medicine and Medical Center – University of Freiburg, Freiburg, Germany; [5]Departments of Biology and Physics, Wilkes University, Wilkes Barre, United States; [6]Gene Editing Group, Institute of Biochemistry II, Goethe University, Frankfurt, Germany; [7]Frankfurt Cancer Institute, Frankfurt, Germany; [8]Institute of Immunology, University Medical Center of the Johannes Gutenberg University Mainz, Mainz, Germany; [9]Department of Hematology, Medical Oncology, & Pneumology, University Medical Center of the Johannes Gutenberg University Mainz, Mainz, Germany; [10]Department of General, Visceral- and Transplant Surgery, University Medical Center, Mainz, Germany; [11]University Cancer Center Mainz, University Medical Center Mainz, Mainz, Germany

**\*For correspondence:**
krishna@uni-mainz.de

**Competing interests:** The authors declare that no competing interests exist.

**Abstract** ERK3 is a ubiquitously expressed member of the atypical mitogen activated protein kinases (MAPKs) and the physiological significance of its short half-life remains unclear. By employing gastrointestinal 3D organoids, we detect that ERK3 protein levels steadily decrease during epithelial differentiation. ERK3 is not required for 3D growth of human gastric epithelium. However, ERK3 is stabilized and activated in tumorigenic cells, but deteriorates over time in primary cells in response to lipopolysaccharide (LPS). ERK3 is necessary for production of several cellular factors including interleukin-8 (IL-8), in both, normal and tumorigenic cells. Particularly, ERK3 is critical for AP-1 signaling through its interaction and regulation of c-Jun protein. The secretome of ERK3-deficient cells is defective in chemotaxis of neutrophils and monocytes both in vitro and in vivo. Further, knockdown of ERK3 reduces metastatic potential of invasive breast cancer cells. We unveil an ERK3-mediated regulation of IL-8 and epithelial secretome for chemotaxis.

## Introduction

Kinases constitute the major component of the druggable genome and deregulation in kinase signaling is associated with nearly 400 diseases (*Hopkins and Groom, 2002*; *Melnikova and Golden, 2004*). The FDA has thus far approved 28 kinase inhibitors for targeted therapeutics (*Wu et al., 2016*). Their clinical success in treating a subset of cancers led to enormous interest in understanding the 'underexplored kinome'. Mitogen activated protein kinases (MAPKs) are a class of serine threonine protein kinases that control fundamental processes like proliferation, migration, differentiation

and cell survival (*Raman et al., 2007*). MAPKs are usually activated in a three-tier signaling cascade and the RAS-RAF-MEK1/2-ERK1/2 is the first characterized MAPK cascade that is activated in response to growth factors (*Wellbrock et al., 2004*). As a paradigm, this cascade is triggered by the activation of the RhoGTPase RAS, which in turn activates the proximal MAPKKK (RAF kinase) of the cascade. MAPKKK kinases then phosphorylate and activate a MAPKK (like MEKs), which ultimately transfer phosphate groups to the threonine or tyrosine residues in the activation segment of the kinase domain of the MAPKs. Classical MAPKs are primarily recognized by the presence of the classical (Thr-X-Tyr) motif, which is phosphorylated by dual specific MAPKKs like MEK1 and MEK2. The activated MAPKs phosphorylate serine and threonine residues in hundreds of substrates, including transcription factors that define a specific response. Among the 14 MAPKs, ERK1, ERK2, p38 α/β/γ/δ and JNK1/2/3 are well characterized.

MAPK6/ERK3 belongs to the unconventional MAPKs that lack the typical Thr-X-Tyr motif in the activation loop of the kinase domain of classical MAPKs (*Cargnello and Roux, 2011*). Despite being ubiquitously expressed, physiological stimuli that trigger ERK3 phosphorylation and activation, as well as its authentic substrates are not well characterized. *ERK3* mRNA is ubiquitously expressed in all tissues with highest expression levels detected in brain, muscles and gastrointestinal tract (*Coulombe and Meloche, 2007*). It was reported that genetic deletion of ERK3 led to a respiratory failure, disturbed growth and neonatal lethality in mice within the first days of life; however, these observations were recently challenged by two publications that confirmed that the observed phenotype was probably attributed to off target effects (*Klinger et al., 2009*; *Ronkina et al., 2019*; *Soulez et al., 2019*). Unlike conventional MAPKs, ERK3 possesses a single phospho-acceptor site at serine 189 within its N-terminus domain, which is constitutively phosphorylated in resting cells. ERK3 is a highly unstable protein with a half-life of 30 to 45 min that undergoes N-terminal ubiquitination. The E3 ubiquitin ligases responsible for this process are currently unknown (*Coulombe and Meloche, 2007*; *Coulombe et al., 2004*; *Coulombe et al., 2003*). However, Usp20 was recently identified as the first deubiquitinating enzyme (DUB) for ERK3 (*Mathien et al., 2017*). The physiological stimuli that activate ERK3 remain unclear and the role of this atypical MAPK in the regulation of physiological processes such as innate immunity has not been studied.

Establishment of stratified epithelium is a fundamental process during development contributing to tissue architecture and it further serves as a first line of defense against pathogen entry in the gastrointestinal tract (*Dotti and Salas, 2018*; *Niessen et al., 2012*; *Otte et al., 2004*). As a constantly differentiating system, gut epithelium is an ideal model to identify genes involved in the process (*Aliaga et al., 1999*). ERK3 has been shown to be required for the maintenance of epithelial architecture; however, genetic deletion studies in mice unveiled a non-essential role for this protein (*Ronkina et al., 2019*; *Takahashi et al., 2018*). To gain further insights into the potential role of ERK3 in epithelial morphogenesis and differentiation, we studied ERK3 expression levels by employing three-dimensional (3D) gastrointestinal organoids that reflect either murine intestinal or human gastric architecture and cell composition (*Almeqdadi et al., 2019*; *Dotti and Salas, 2018*). Epithelial cells respond to innate immune stimuli by secreting key inflammatory molecules to promote chemotaxis of immune cells to the micromilieu (*Li et al., 1998*; *Okumura and Takeda, 2017*; *Onyiah and Colgan, 2016*). IL-8 is one of the first chemokines released by injured or inflamed epithelium, and it is constitutively expressed by intestinal epithelial cells (*Eckmann et al., 1993*). Being a prime modulator of the epithelial immune responses, IL-8 expression is coordinated by multiple signaling pathways (*Jundi and Greene, 2015*; *Wen and Wu, 2001*). Transcriptional activation of the IL-8 promoter strictly relies on the binding of inducible transcription factors (TFs) such as NF-κB and activating protein 1 (AP-1) (*Hoffmann et al., 2002*; *Jundi and Greene, 2015*). The AP-1 transcription factor is a homodimeric/heterodimeric protein complex formed between Jun and Fos proteins (*Jundi and Greene, 2015*; *Leppä and Bohmann, 1999*). Its transcriptional potential is regulated by protein phosphorylation, its abundance and interaction with protein kinases like MAPKs in the nuclei (*Hoffmann et al., 2002*). The AP-1 transcription factor c-Jun can translocate and be retained in the nucleus as a monomer, without binding to its target sequence (*Karin, 1996*; *Schreck et al., 2011*). Although c-Jun is predominantly regulated by JNK it was demonstrated that its nuclear translocation is independent of its phosphorylation by and interaction with JNK (*Schreck et al., 2011*). Here, we unveil a kinase-independent role of ERK3 in the regulation of the epithelial secretome including maintenance and induction of IL-8, in both, normal and tumorigenic cells. ERK3-mediated IL-8 secretion is critical for the chemotaxis of leukocytes to the epithelium. We further unveil a synergistic role

of ERK3 and classical MAPKs in the regulation of epithelial secretome and identify ERK3 as a novel interacting partner of c-Jun and regulator of AP-1 activity.

## Results

### ERK3 protein dynamics during morphogenesis and differentiation of gastrointestinal organoids

To elucidate the physiological role of ERK3 in the establishment of stratified epithelium, we generated murine and human gastrointestinal organoid cultures as described in the Materials and methods. Undifferentiated mouse colon organoids (MCOs) and patient-derived human gastric organoids (HGOs) were expanded and epithelial differentiation was induced by withdrawal of Wnt3A and R-spondin 1 (RSP1) from the media. We evaluated the expression of *ERK3* mRNA and protein levels in the differentiating organoids (*Figure 1A*). The differentiation of these organoids was confirmed by the expression of epithelial differentiation markers including Keratin 20 (Krt20). As expected, induction of differentiation led to a high expression of Krt20 in differentiated organoids (*Figure 1B*, *Figure 1—figure supplement 1A*). Expression levels of ERK3 protein and mRNA were analyzed by western blotting and real-time PCR (RT-PCR). These data revealed that ERK3 protein levels are significantly lower in the differentiated MCOs as compared to the undifferentiated ones (*Figure 1C–D*). The mRNA levels of *Erk3* were slightly higher under differentiated conditions (*Figure 1E*). Taken together, these data suggest that the decrease in ERK3 expression observed in the differentiated MCOs is regulated at the protein levels.

We then performed similar experiments in HGOs which recapitulated the ERK3 expression pattern observed in the MCOs. ERK3 protein levels steadily decrease during the differentiation of HGOs without much change in the mRNA levels (*Figure 1F–I*, *Figure 1—figure supplement 1C*). In conclusion, these data indicate that ERK3 proteostasis might play a role in epithelial differentiation and function.

### LPS exerts opposing effects on ERK3 proteostasis in primary epithelial cells and colon carcinoma cell lines

Epithelial monolayer serves as the first line of defense against pathogens in the gastrointestinal tract; however, despite being constantly exposed to significant levels of luminal LPS, the primary epithelium does not display inflammation under physiological conditions (*Otte et al., 2004*).

To investigate the activation dynamics of ERK3 in response to LPS in human epithelial cells, we challenged colorectal adenocarcinoma HT-29 cells with LPS for various time points as described in the Materials and methods. Interestingly, LPS stimulation led to a time-dependent increase in the protein levels of ERK3 with concomitant increase in the phosphorylation status of S189 during the time frame of the experiment (*Figure 2A and B*). We did not detect any changes in the protein levels of the closely related MAPK ERK4 (MAPK4) (*Figure 2A*). MK5 (MAPKAPK5), one of the proposed downstream substrates of ERK3 and ERK4 was also activated in response to LPS, as revealed by the increased T182 phosphorylation of MK5 in LPS-treated HT-29 cells (*Figure 2A and C*). Similarly, we also detected an increase in pERK3 levels in colorectal adenocarcinoma CaCo2 cells in response to LPS (*Figure 2—figure supplement 1*). Further, expression levels of *ERK3* mRNA were determined with quantitative RT-PCR with no significant increase detected upon LPS treatment (*Figure 2D*). RT-PCR analyses coupled with cycloheximide (CHX) chase experiments revealed that LPS upregulates ERK3 protein levels (*Figure 2E*), while not exerting any significant increase in ERK3 protein half-life (*Figure 2F*). These data suggested that ERK3 might possibly play a role in LPS-mediated responses in the epithelium.

We then tested if the observed effects are confined to tumorigenic cells by performing similar experiments in human colonic primary epithelial cells (HCPECs). Interestingly, in these cells, LPS stimulation led to a decrease in the protein levels of ERK3, while activating MK5 and the classical MAPKs: ERK1/2 and p38 (*Figure 2G–K*). These results also indicate that MK5 might be phosphorylated and activated by p38 MAPK or the closely related ERK4 as these two MAPKs have also been shown to phosphorylate MK5 at T182 (*Aberg et al., 2009*; *New et al., 1998*). Interestingly, LPS stimulation also affects the total protein levels of MK5 in a kinetic manner (*Figure 2G and I*). It has been reported that unlike the tumorigenic cells, primary epithelial cells express low levels of Toll-like

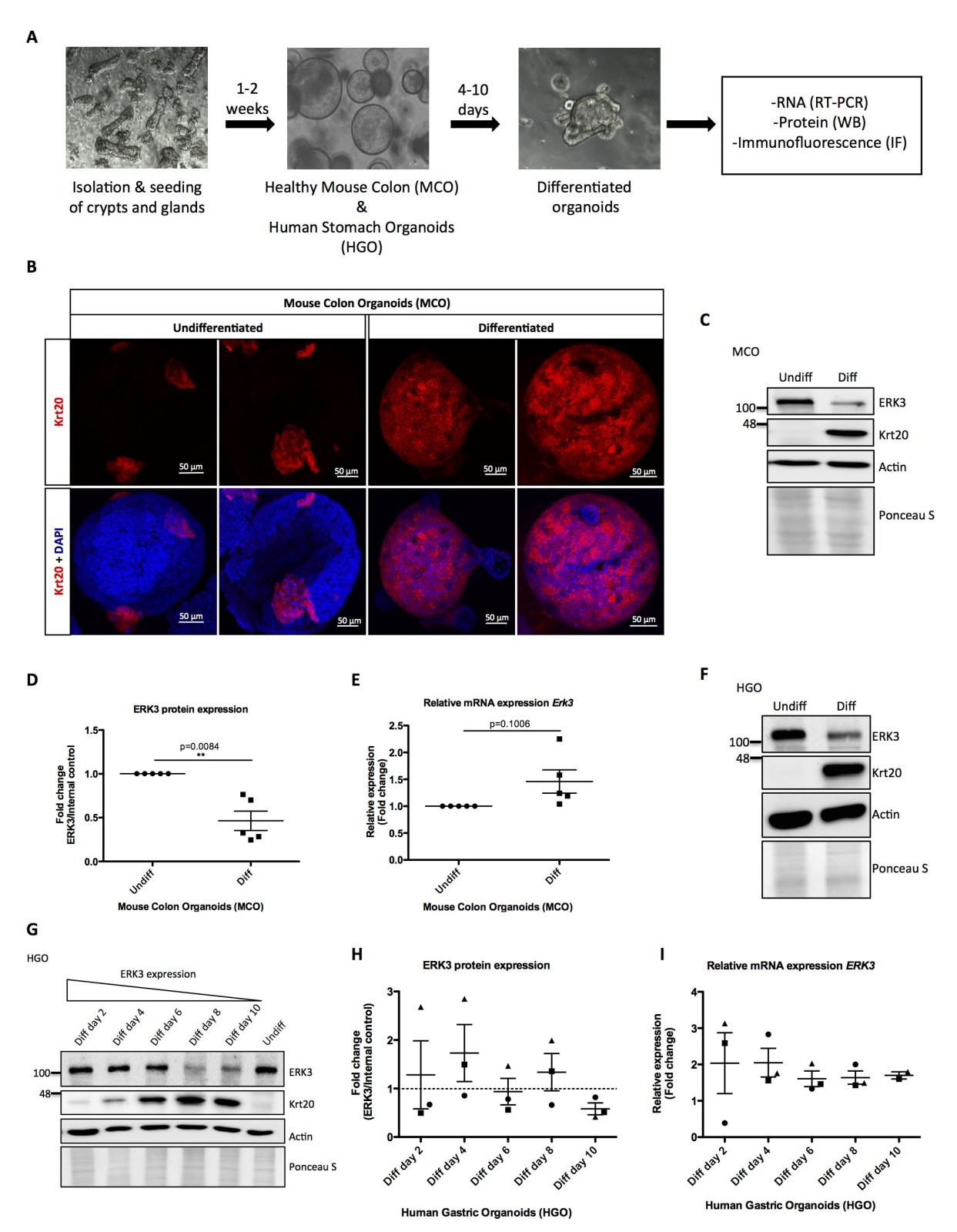

**Figure 1.** Role of ERK3 in gastrointestinal organoids morphogenesis and differentiation. (**A**) Schematic outline of individual steps for organoid establishment and differentiation. Undifferentiated, healthy mouse colon organoids (MCO) or human gastric organoids (HGO) were differentiated by withdrawal of Wnt3A and RSP1 and subjected to western blot (WB), RT-PCR and/or immunofluorescence staining (IF) analyses. (**B**) Representative micrographs of undifferentiated and differentiated MCO. Shown are stainings with differentiation marker Keratin (20) (Krt20) (red) and DAPI (blue). Scale

*Figure 1 continued on next page*

*Figure 1 continued*

bar 50 µm. (C) Representative immunoblot analysis of undifferentiated (Undiff) and differentiated (Diff) MCO. MCO were seeded in matrigel, 3 days post-seeding differentiation process was induced. Organoids were lysed on day 7 for WB analyses using antibodies against total ERK3 and Krt20 differentiation marker. Actin and Ponceau S staining were used as loading controls. (D) Fold change in ERK3 protein expression presented as a ratio Diff/Undiff after normalization with the internal loading control (Actin/Ponceau S). Data derived from five independent experiments (n = 5) are represented as mean ± SEM fold change; *p<0.05, **p<0.01, ***p<0.001, paired t-test. (E) Quantitative RT-PCR analysis of *Erk3* expression in differentiated organoids when compared to undifferentiated organoids. Each biological replicate was measured in triplicates. Log2 fold change in gene expression is presented as mean ± SEM of five independent experiments (n = 5); *p<0.05, **p<0.01, ***p<0.001, paired t-test. Expression of differentiation markers: *Krt20* and intestinal alkaline phosphatase (*Alpi*) (enterocyte marker) was determined by RT-PCR and is presented in *Figure 1— figure supplement 1A-B*. (F) WB analysis of HGOs under undifferentiated and differentiated conditions. HGOs were seeded in matrigel and after 4 days differentiation was started by withdrawal of Wnt3A and RSP1. Organoids were lysed on day 10 and levels of ERK3 and Krt20 were assessed by WB analysis. Actin and Ponceau S staining were used as loading controls. (G) Representative western blot analysis of ERK3 kinetics in HGOs upon differentiation. HGOs were seeded in matrigel and differentiation was induced 4 days post-seeding. Organoids were lysed on days 2, 4, 6, 8 and 10, levels of ERK3 were monitored. Krt20 expression was used as a differentiation marker and actin/Ponceau S staining as loading controls. (H) ERK3 expression in differentiating HGO was calculated in respect to the undifferentiated organoids after normalization with loading control and is presented as mean fold change ± SEM from three biological replicates of HGOs (n = 3) from two different patients. (I) Relative expression of *ERK3* was assessed by RT-PCR in differentiated HGOs (from two different patients) on days 2, 4, 6, 8 and 10 in respect to the undifferentiated organoids and is presented as mean log2 fold change ± SEM from three independent experiments (n = 3) except of day 10 (Diff day 10) where two biological replicates are depicted (n = 2). Expression of the Gastrokine 1 (*GKN1*) differentiation marker was monitored and is presented in *Figure 1—figure supplement 1C*. The online version of this article includes the following source data and figure supplement(s) for figure 1:

**Source data 1.** Full membrane scans for western blot images for *Figure 1C, F and G*.

**Figure supplement 1.** Differentiation of mouse and human gastrointestinal organoids.

receptor 4 (TLR4) (*Cario and Podolsky, 2000*; *Pott and Hornef, 2012*). This observation was confirmed in our cell culture models for HCPECs and HT-29 cells (*Figure 2—figure supplement 2*). Despite the observed discrepancy in primary and oncogenic cells in response to LPS, the effects observed are attributed to the altered proteostasis of ERK3, as *ERK3* mRNA levels are not significantly altered in response to LPS in both cell types (*Figure 2D and L*). Consistently, the polyubiquitination of endogenous ERK3 is modulated in response to LPS in both HT-29 cells and HCPEC: LPS treatment repressed and induced ERK3 ubiquitination in oncogenic and primary epithelial cells, respectively (*Figure 2M and N*, respectively). These results confirmed that LPS-mediated turnover of ERK3 protein is partially mediated via the ubiquitin proteasome machinery.

We then tested whether ERK3 is activated in response to other innate immune stimuli like TLR1/2 ligand Pam3CSK4, TLR7/8 ligand R848 and IL-1β. We detected an increase in the phosphorylation of ERK3 (S189) in response to IL-1β and Pam3CSK4, in both, HCPECs (*Figure 2—figure supplement 3A–3B*) and HT-29 cells (*Figure 2—figure supplement 3D–3E*). In contrast to the other stimuli, we detected a decrease in S189 phosphorylation with TLR7/8 ligand R848 in both cell types (*Figure 2— figure supplement 3C and F*). These data suggest that ERK3 is preferentially activated in a cell type- and stimulus-dependent manner.

## ERK3 regulates transcription of *CXCL8* and other key chemotactic factors

To understand the physiological relevance of ERK3 in modulating LPS-mediated innate immune responses in the epithelium, we performed RNA sequencing analysis in the presence and absence of ERK3. HCPECs wild type (WT) (siCo) or ERK3-depleted (siERK3) were challenged with LPS for 24 hr and differentially expressed (DE) factors were analyzed (*Figure 3* and *Figure 3—source data 1*). Several key cytokines and chemokines contributing to epithelial function were found to be downregulated in ERK3-depleted cells, suggesting that ERK3 is probably required for the positive regulation of the epithelial secretome (*Figure 3D* and *Figure 3—source data 1*). To further evaluate these observations, we performed a direct secretome analysis of control vs ERK3-depleted HCPECs as described in the Materials and methods (*Figure 3- Figure 3—source data 2*). Interestingly, we saw that many of the differentially regulated factors identified in the transcriptome analysis were also altered at the protein levels (*Figure 3- Figure 3—source data 3*). In particular, we discovered that ERK3 is required for the transcriptional regulation of *CXCL8*, *CXCL10*, *CCL2* and *CXCL6* in HCPECs (*Figure 4A*). Indeed, we validated that ERK3 regulates the mRNA levels of *CXCL8*, *IL16* and *CXCL6* in HCPECs (*Figure 4—figure supplement 1A–C*). Further, gene ontology (GO) enrichment analysis

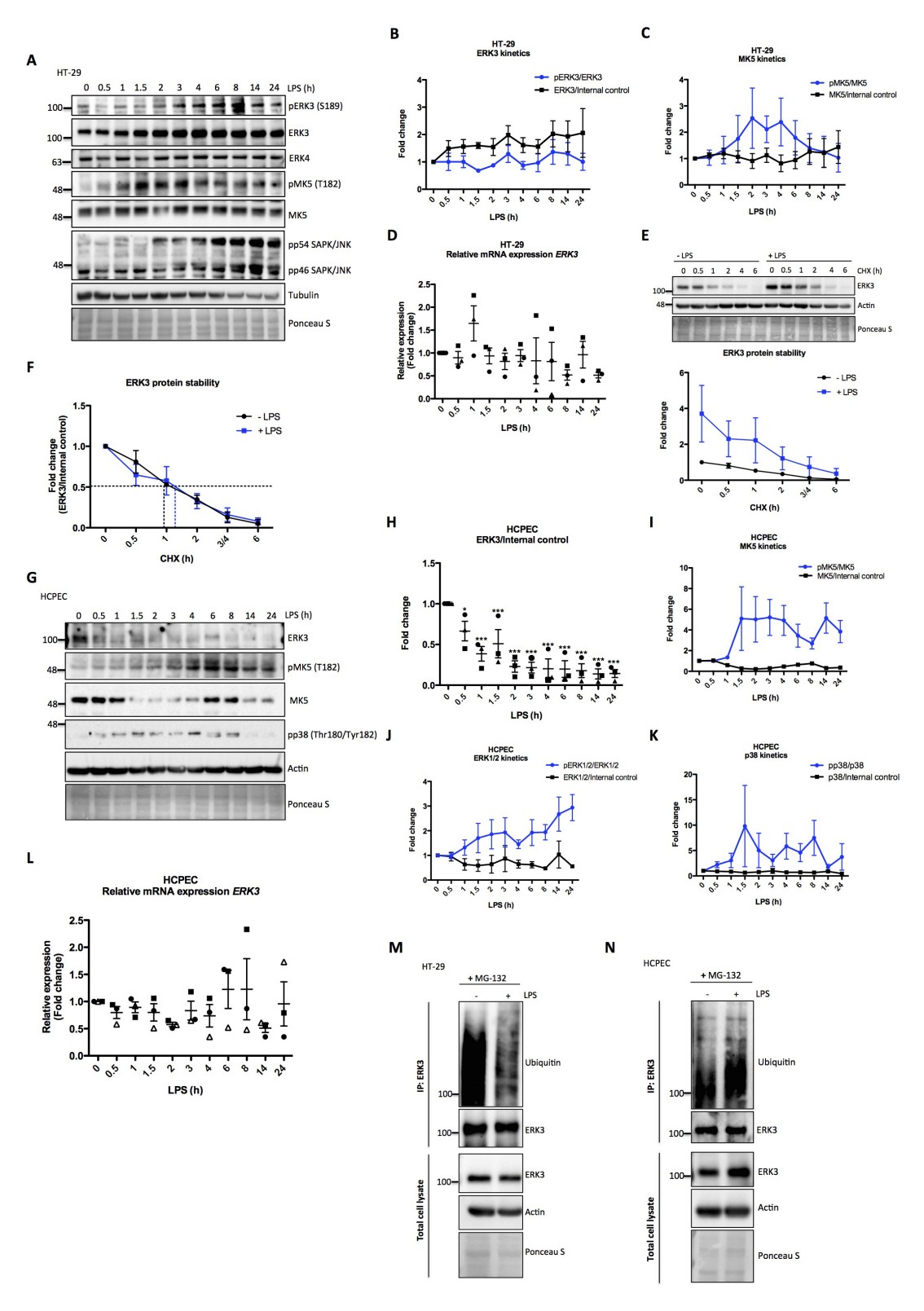

**Figure 2.** Effect of LPS on ERK3 expression and protein stability. HT-29 cells and HCPECs were stimulated with LPS (200 ng/ml) at various time points as indicated. (**A**) Representative western blot analysis of HT-29 cells. Phosphorylation and/or total protein levels of ERK3, ERK4, MK5 and JNK were monitored. Tubulin immunoblots and Ponceau S staining were employed to monitor equal loading. (**B**) Changes in the expression and phosphorylation of ERK3 protein are shown as a fold change after normalization with internal loading control. Each time point was normalized to unstimulated cells (0).

*Figure 2 continued on next page*

*Figure 2 continued*

Fold change values from three independent experiments (n = 3) are represented as mean ± SEM. (C) Activation kinetics of MK5 in HT-29 cells stimulated with LPS. Fold change in MK5 phosphorylation levels upon LPS stimulation normalized to total protein levels and expression levels of MK5 normalized to internal loading control are shown. Fold change values from three independent experiments (n = 3) are represented as mean ± SEM. (D) Quantitative RT-PCR analysis of *ERK3* expression. Each biological replicate was measured in triplicates. Log2 fold change in gene expression is presented as mean ± SEM of three independent experiments (n = 3); *p<0.05, **p<0.01, ***p<0.001, one-way ANOVA, Turkey's post-test. (E) ERK3 protein stability was assessed by CHX chase at 0 hr, 0.5 hr, 1 hr, 2 hr, 3/4 hr and 6 hr in the presence and absence of LPS (30 min pre-treatment). Western blot analyses were performed and representative results are presented. ERK3 protein levels in respect to the untreated cells (-LPS, 0 hr) were calculated using ImageJ and data are presented as mean fold change ± SEM from three independent experiments (n = 3). (F) Graph presents ERK3 protein levels quantified in respect to the untreated cells (0) of unstimulated (-LPS) and LPS stimulated (+LPS) cells, respectively and data are presented as mean fold change ± SEM from three independent experiments (n = 3). (G) HCPECs were stimulated with LPS and immunoblot analyses of the phosphorylation and/or total protein levels of ERK3, MK5 and p38 were performed. Actin and Ponceau S staining were used as loading controls. (H) Plotted here are fold changes in expression of ERK3 protein. Results are shown as mean ± SEM fold change after normalization with the levels of internal loading control. Each time point was normalized in respect to unstimulated HCPECs (0). Data are a representative of three independent experiments (n = 3); *p<0.05, **p<0.01, ***p<0.001, one-way ANOVA, Turkey's post-test. (I-K) Plotted here are fold changes in the phosphorylation of (I) MK5 at T182, (J) ERK1/2 and (K) p38 in response to LPS stimulation normalized to the respective total protein levels as well as the expression levels of total proteins normalized in respect to the internal loading control. Each time point was normalized in respect to the unstimulated cells (0). Fold change values are presented as mean ± SEM from three independent experiments (n = 3). (L) Quantitative RT-PCR analysis of *ERK3* mRNA expression levels. Log2 fold change in gene expression is presented as mean ± SEM of three independent experiments (n = 3); *p<0.05, **p<0.01, ***p<0.001, one-way ANOVA, Turkey's post-test. (M-N) LPS-mediated ubiquitination of endogenous ERK3 in (M) HT-29 cells and (N) HCPECs. HT-29 cells and HCPECs were seeded and treated as mentioned in the Materials and methods. Total cell lysates (TCL) and endogenous ERK3 immunoprecipitates (IP) were analyzed by immunoblotting. Levels of ERK3 and polyubiquitination were monitored. Actin and Ponceau S staining were used as loading controls for TCL western blot analysis. Results are representatives of at least two experiments showing the same tendency. ERK3 kinetics in response to other immune stimuli are presented in *Figure 2—figure supplement 3*).

The online version of this article includes the following source data and figure supplement(s) for figure 2:

**Source data 1.** Full membrane scans for western blot images for *Figure 2A, E, G, M and N*.
**Figure supplement 1.** ERK3 kinetics in CaCo2 cells.
**Figure supplement 1—source data 1.** Full membrane scans for western blot images for *Figure 2—figure supplement 1A*.
**Figure supplement 2.** Graph representing relative mRNA expression levels of *TLR4* in HCPECs when compared with HT-29 cells.
**Figure supplement 2—source data 1.** Full membrane scans for western blot images for *Figure 2—figure supplement 2A–F*.
**Figure supplement 3.** ERK3 kinetics in HCPECs and HT-29 in response to IL-1β, TLR1/2 ligand Pam3CSK4 and TLR7/8 ligand R848.

was performed on the DE genes associated with the regulation of chemotaxis and immune responses to determine biological processes that might be regulated by ERK3 in HCPECs. Selected pathways ranked in accordance to p-value eliminated are presented as a table in *Figure 3—figure supplement 1*. A heatmap for the genes associated with positive regulation of leukocyte chemotaxis was generated and differential expression of the relevant genes is presented for control and ERK3-depleted HCPECs (*Figure 3—figure supplement 1*). These analyses confirmed that ERK3 plays an important role in the regulation of genes involved in, among others, immune responses and leukocyte chemotaxis.

## IL-8 protein levels reflect ERK3 expression status in LPS stimulated HCPECs and HT-29 cells

Considering that ERK3 regulates IL-8 production and the fact that LPS treatment exerts an opposing effect on ERK3 protein levels in HCPECs and oncogenic HT-29 cells (*Figure 2A–G*, respectively), levels of CXCL8/IL-8 were determined in both cell lines in response to LPS. Interestingly, while IL-8 levels increased over time in HT-29 cells, drastic reduction in chemokine levels was observed in HCPECs upon LPS stimulation (*Figure 4B and C*, respectively). More interestingly, IL-8 levels measured in HT-29 and HCPECs in response to LPS reflected ERK3 expression status in both cell types (*Figure 4B–C*, *Figure 2A–B*, and *Figure 2G–H*). In contrast to cancer cells, HCPECs display a high basal level of IL-8 (*Figure 4B and C*). Depletion of ERK3 with multiple siRNAs and shRNAs led to a significant reduction in both, basal as well as LPS-induced IL-8 levels, suggesting an obligatory role for ERK3 in the maintenance of IL-8 not only in HCPECs and HT-29 cells, but also in MDA-MB231 and HeLa cells (*Figure 4D–F* and *Figure 4—figure supplement 2*). To further corroborate these observations, we established ERK3 knockout cells by employing a CRISPR/Cas9 approach. Consistent with the observations made with si- and shRNAs, we affirmed that CRISPR/Cas9 knockout of

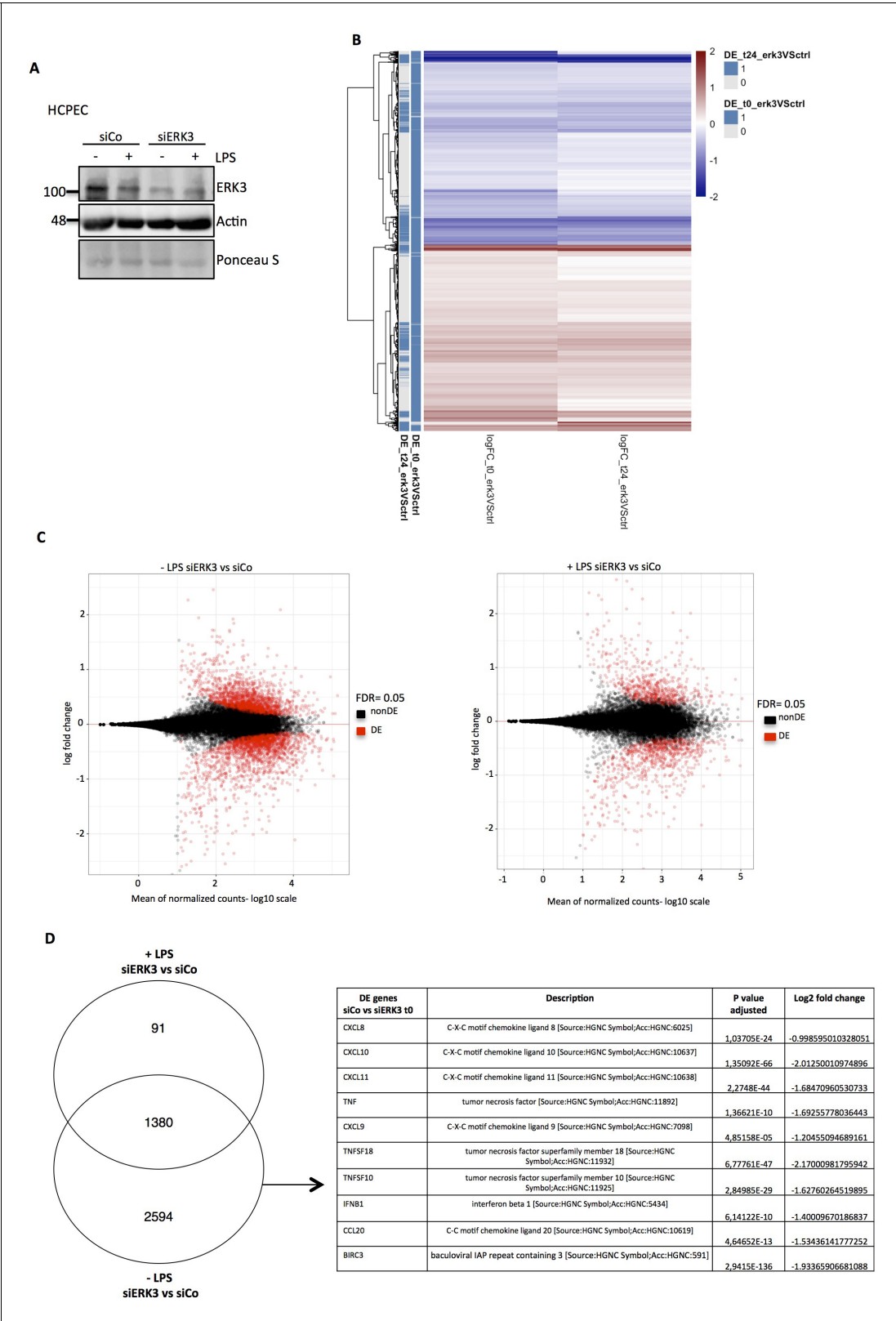

**Figure 3.** RNA sequencing analysis of control and ERK3 knockdown HCPECs. HCPECs were transiently transfected with either control siRNA (siCo) or siRNA targeting ERK3 (siERK3). 24 hr post-transfection medium was exchanged and cells were stimulated with LPS. 24 hr later supernatants were harvested for each variant for further secretome analysis and the isolated RNA was directed for RNAseq analysis. (**A**) Representative western blot analysis of control (siCo) and ERK3-depleted (siERK3) HCPECs under -/+ LPS conditions subjected for RNAseq and secretome analyses. Levels of ERK3

*Figure 3 continued on next page*

*Figure 3 continued*

were determined to estimate knockdown efficiency, actin and Ponceau S are provided as loading controls. (B) Heatmap representing differentially expressed (DE) genes. (C) DE genes in resting and LPS stimulated siERK3 cells as compared to siCo cells. (D) Visual representation of DE genes in siERK3 cells when compared with siCo cells. Table represents DE genes downregulated upon ERK3 depletion in unstimulated (-LPS) HCPECs.
The online version of this article includes the following source data and figure supplement(s) for figure 3:

**Source data 1.** RNA sequencing analysis of control and ERK3 knockdown HCPECs.
**Source data 2.** Secretome analysis of control and ERK3-depleted HCPECs.
**Source data 3.** Combined transcriptome and secretome analysis of control and ERK3-depleted HCPECs.
**Source data 4.** Full membrane scans for western blot images for *Figure 3A*.
**Figure supplement 1.** RNAseq analysis: TopGO functional interpretation of the DE genes in siERK3 HCPECs.

ERK3 reduces LPS-induced IL-8 levels in CRISPR ERK3 HT-29 cells (*Figure 4G–H*). Furthermore, we performed loss-of-function studies in HGOs by employing lentivirus-based CRISPR/Cas9 system. The infected cells were then selected for antibiotic resistance and the efficiency of the knockout was verified by western blots. Data obtained from HGOs of two different patients revealed that ERK3 is not required for the 3D growth of human gastric epithelium as we failed to detect any significant change in the number or the size between the control (CRISPR Co) and ERK3-depleted (CRISPR ERK3) HGOs (*Figure 4I–J*). However, consequently with the observations made for HCPECs, knockout of ERK3 impaired IL-8 secretion in human gastric organoids from two different patients (*Figure 4K*). We also verified mRNA expression of *CXCL8* in HGOs upon differentiation (*Figure 4L*), which correlates with the expression pattern of ERK3 protein levels (*Figure 1G–H*).

Next, we investigated the role of ERK3 in the production of IL-8 stimulated by IL-1β and TLR1/2 ligand Pam3CSK4, the other two ERK3 activating stimuli (*Figure 2—figure supplement 3*) apart from LPS. 24 hr stimulation with IL-1β induced ERK3 protein expression in control (siCo) HCPECs with concomitant upregulation of IL-8 (*Figure 4—figure supplement 3A–3B*). However, knockdown of ERK3 (siERK3) did not reduce IL-8 levels induced by IL-1β (*Figure 4—figure supplement 3A–3B*) suggesting that activation of p38 MAPK, but not ERK3 is probably contributing to IL-8 production in primary intestinal epithelial cells as shown before (*Parhar et al., 2003*). On the contrary, depletion of ERK3 inhibited the induction of IL-8 upon stimulation of these cells with Pam3CSK4 despite the activation of p38 MAPK (*Figure 4—figure supplement 3C–3D*).

Unlike in primary cells, stimulation of HT-29 cells with IL-1β induced IL-8 in an ERK3-dependent manner (*Figure 4—figure supplement 3E–3F*). However, stimulation of HT-29 with Pam3CSK4 failed to induce IL-8 production, irrespective of the presence or absence of ERK3 protein and activation of p38 MAPK (*Figure 4—figure supplement 3G–3H*).

## Cross-talk between canonical MAPKs and ERK3 in the regulation of IL-8

Activity of ERK1/2 has been implicated to be crucial for epithelial cell differentiation (*Aliaga et al., 1999*). Moreover, in several experiments, we found that despite the activation of ERK1/2, direct (ERK3 depletion) and indirect (LPS stimulation in HCPECs) downregulation of ERK3 negatively affected IL-8 production. To further explore the role of canonical MAPKs in the regulation of epithelial cell function, we checked for a potential cross-talk between ERK1/2 and ERK3 in controlling IL-8 levels, as several studies unveiled a critical role for classical MAPKs in the production and secretion of CXCL8/IL-8 (*Hartman et al., 2017*; *Lee et al., 2006*; *Marie et al., 1999*). Most of the studies investigating the role of ERK1/2 were employing the MEK1 inhibitor trametinib to evaluate the role of this pathway in mediating several cellular processes. Blocking of the MEK1/2-ERK1/2 pathway by trametinib led to a significant reduction of LPS-induced IL-8 levels in HT-29 cells (*Figure 4—figure supplement 4A–B*). Interestingly, we observed that treatment with trametinib severely compromised protein levels of ERK3 in HT-29 cells (*Figure 4—figure supplement 4A–B*) with no significant effect on the *ERK3* mRNA levels (*Figure 4—figure supplement 4C*). Since ERK3 protein abundance played a critical role in IL-8 regulation and trametinib induced the downregulation of ERK3 protein levels, we tested whether the IL-8 decrease observed upon ERK1/2 inhibition was induced by blocked activity of the MEK1/2-ERK1/2 module or by an indirect attenuation of ERK3 protein abundance. To address this issue, HT-29 cells were pre-treated with the proteasome inhibitor MG-132 to prevent proteasomal degradation, prior to trametinib treatment which was followed by LPS stimulation to induce IL-8 production. Systemic block in proteasomal activity with MG-132 led to the accumulation

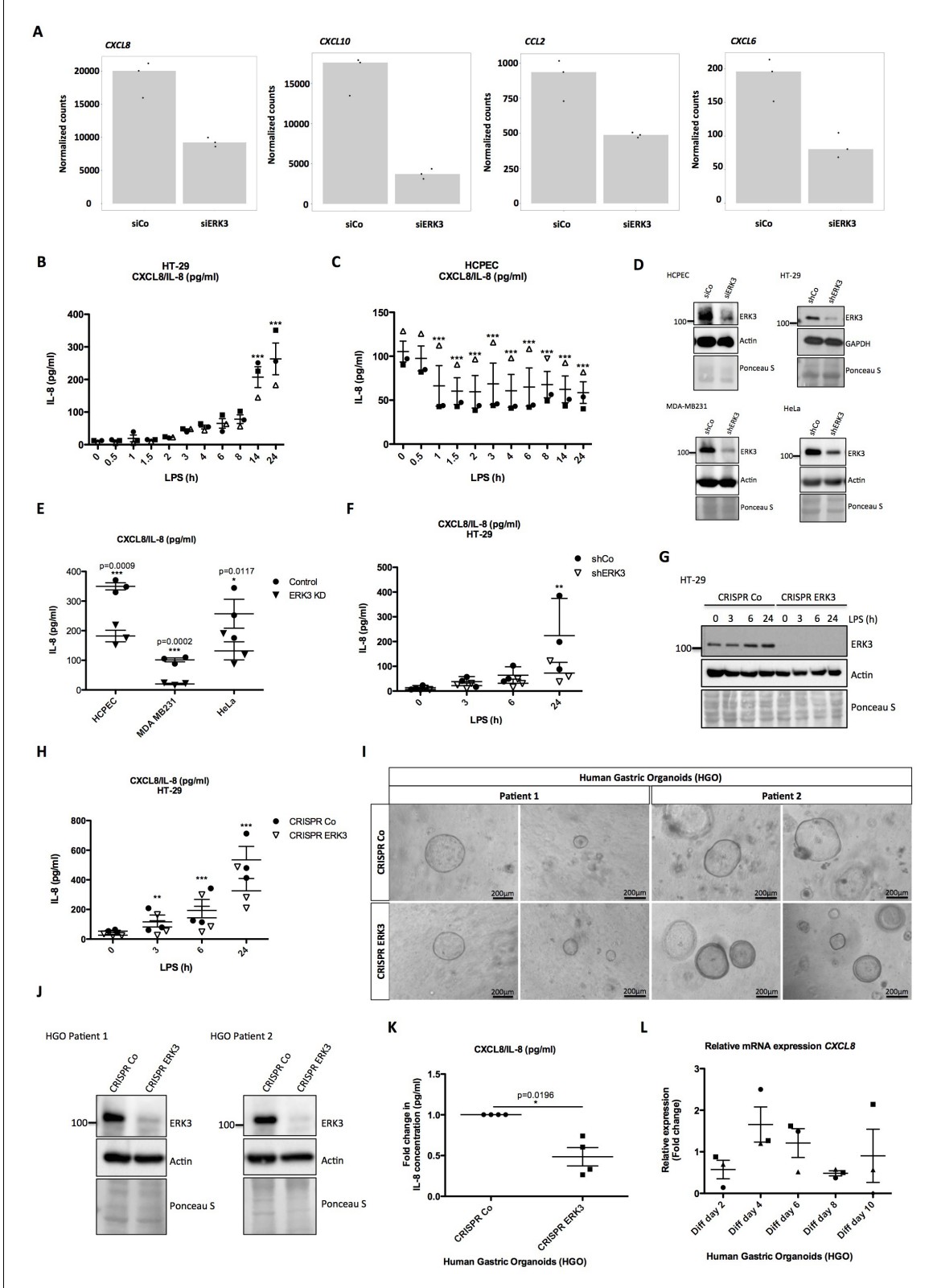

**Figure 4.** ERK3 regulates levels of IL-8 and other chemokines. (**A**) Graphs represent RNAseq-derived normalized counts between siControl and siERK3 samples from three biological replicates of unstimulated HCPECs. (**B-C**) ELISA was performed to determine CXCL8/IL-8 levels in (**B**) HT-29 and (**C**) HCPECs in response to LPS. IL-8 concentrations presented as mean ± SEM were obtained from three independent experiments (n = 3); *p<0.05, **p<0.01, ***p<0.001, one way ANOVA, Turkey's post-test. (**D-F**) Secretion of CXCL8/IL-8 was measured in supernatants obtained from different cell

*Figure 4 continued on next page*

Figure 4 continued

lines. Each cancer cell line was stably transduced with lentiviral particles carrying either shRNA empty vector control or with shRNA targeting ERK3. HCPECs were transiently transfected with either control siRNA or siRNA targeting ERK3. After 24 hr, medium was exchanged for each cell line for serum and supplements free medium and cells were cultured for 48 hr prior supernatant collection. (D) All cell lines were subjected to western blot analysis and ERK3 knockdown efficiency was estimated. Ponceau S staining and actin/GAPDH loading controls are provided. Depicted western blots are representative for at least three independent experiments (n = 3). (E) IL-8 levels were measured by ELISA. Shown are data representing mean ± SEM concentration in pg/ml from three biological replicates (n = 3); *p<0.05, **p<0.01, ***p<0.001, t-test. (F) Control (shCo) or ERK3 knockdown HT-29 cells (shERK3) presented in D were stimulated with LPS at indicated time points. Supernatants were harvested for each time point and IL-8 levels were measured by ELISA. Data are representing mean concentration in pg/ml ± SEM from three biological replicates per condition (n = 3); *p<0.05, **p<0.01, ***p<0.001, two-way ANOVA, Bonferroni post-tests. (G-H) Control (CRISPR Co) and ERK3 knockout (CRISPR ERK3) HT-29 cells were stimulated with LPS for 0 hr, 3 hr, 6 hr or 24 hr, supernatants were harvested and IL-8 levels were monitored. (G) Cells were analyzed by immunoblotting, levels of ERK3 expression were monitored. Ponceau S staining and actin are included as loading controls. Data are a representative of three biological experiments (n = 3). (H) IL-8 concentrations in pg/ml are presented as mean ± SEM obtained from three independent experiments (n = 3); *p<0.05, **p<0.01, ***p<0.001, two-way ANOVA, Bonferroni post-tests. Additionally, IL-1β- and Pam3CSK4-dependent regulation of IL-8 was tested in control and ERK3-depleted HCPECs and HT-29 cells and data are presented in *Figure 4—figure supplement 3*. Cross-talk between ERK1/2 and ERK3 is presented in *Figure 4—figure supplement 4*. (I) Phase-contrast microscopy control (CRISPR Co) and CRISPR ERK3 HGOs from two patients. Organoids were dissociated into single cells and transduced with either CRISPR control (CRISPR Co) or CRISPR ERK3 lentiviral particle-containing supernatants. 4 hr post-infection, cells were seeded in matrigel and 48 hr later 3D cultures were selected with 5 µg/ml of puromycin. Organoids were split and cells were collected for immunoblot analyses. Bright field images of both cultures were taken on day seven after splitting. Scale bar represents 200 µm. (J) Western blot analysis of control (CRISPR Co) and ERK3 knockout (CRISPR ERK3) HGOs. ERK3 expression was monitored to verify knockdown efficiency, actin and Ponceau S staining are provided as loading controls. (K) IL-8 protein secretion was measured in supernatants obtained from control (CRISPR Co) and ERK3 knockout (CRISPR ERK3) HGOs from two patients. Supernatants were harvested from two different 3D organoid cultures after two rounds of puromycin-based selection and after splitting of the organoids. Mean fold change in IL-8 concentrations (pg/ml)± SEM is presented for CRISPR ERK3 supernatants in respect to the control samples (CRISPR Co) obtained from four biological replicates (n = 4); *p<0.05, **p<0.01, ***p<0.001, paired t-test. (L) Relative expression of *CXCL8* mRNA was measured at different days of HGOs differentiation. Presented are mean fold changes ± SEM in *CXCL8* expression in differentiated organoids normalized in respect to the undifferentiated HGOs from three different cultures of gastric organoids for two patients. Data represent three biological replicates (n = 3), except of day 10 (Diff day 10) where two biological replicates are depicted (n = 2). *p<0.05, **p<0.01, ***p<0.001, one way ANOVA, Turkey's post-test.

The online version of this article includes the following source data and figure supplement(s) for figure 4:

**Source data 1.** Full membrane scans for western blot images for *Figure 4D, G and J*.
**Figure supplement 1.** Validation of select chemokines and cytokines by RT-PCR.
**Figure supplement 2.** Validation of ERK3-IL-8 connection by different shRNAs and siRNAs.
**Figure supplement 2—source data 1.** Full membrane scans for western blot images for *Figure 4—figure supplement 2A and B*.
**Figure supplement 3.** Role of ERK3 in the production of IL-8 stimulated by IL-1β and Pam3CSK4.
**Figure supplement 3—source data 1.** Full membrane scans for western blot images for *Figure 4—figure supplement 3A, C, E and G*.
**Figure supplement 4.** Trametinib treatment leads to a decrease in IL-8 secretion and ERK3 protein expression.
**Figure supplement 4—source data 1.** Full membrane scans for western blot images for *Figure 4—figure supplement 4A, B and D*.
**Figure supplement 5.** Role of MK5 and ERK4 in the regulation of ERK3 and IL-8.
**Figure supplement 5—source data 1.** Full membrane scans for western blot images for *Figure 4—figure supplement 5A and C*.

of ERK3 protein and rescued IL-8 levels decreased by MEK inhibitor treatment in both, resting and LPS stimulated HT-29 cells despite inactive ERK1/2 (*Figure 4—figure supplement 4D and E*). Interestingly, MG-132 treatment itself caused an increase in IL-8 levels comparable with the upregulation obtained by LPS stimulation, while no significant effect was detected in ERK1/2 phosphorylation (*Figure 4—figure supplement 4D and E*). These data suggest that proteasomal activity is required for the regulation of IL-8 levels and that ERK3 turnover via the proteasomes potentially contributes toward this phenotype in cancer cells.

## Functional interplay between ERK4 and MK5 in the regulation of ERK3 and IL-8

We also evaluated the role of ERK4 and MK5, the other related components of the ERK3 signaling module. Our observations indicate while endogenous MK5 positively regulates ERK3 expression, ERK4 exerts opposing effects on ERK3 protein levels (*Figure 4—figure supplement 5*). As MK5 and ERK4 exert differential effects on ERK3 protein levels, we checked for IL-8 levels in these cells. We observed that IL-8 levels correlated with ERK3 protein levels in MK5- and ERK4-depleted cells (*Figure 4—figure supplement 5A–E*).

## ERK3-mediated regulation of IL-8 is kinase independent

We then investigated if the kinase activity of ERK3 is required for the production and secretion of IL-8. To address this issue, we performed complementation experiments by expressing exogenous ERK3 (WT) and kinase-dead ERK3 (K49A/K50A) in shERK3 cells where the endogenous ERK3 expression is suppressed by an shRNA targeted to the 3'UTR region of the mRNA. Interestingly, expression of WT as well as the kinase-dead version rescued the secretion of IL-8 in at least two different cell lines (*Figure 5A–B*). Taken together, these results suggested a kinase-independent role for ERK3 in the regulation of IL-8 levels.

## Depletion of ERK3 reduces metastatic potential of breast cancer cell line MDA-MB231

Next, we evaluated the role of the ERK3-IL-8 axis under pathological conditions like cancer metastasis. Role of IL-8 in tumor progression has been well established (*Chan et al., 2017*; *Long et al., 2016*), and high levels of this chemokine are a poor prognostic factor in melanoma, breast, liver, lung and colon cancer (*David et al., 2016*; *Ueda et al., 1994*). Previous studies on ERK3 indicated its role in tumorigenesis, including regulation of migratory properties of MDA-MB231 breast cancer cells (*Al-Mahdi et al., 2015*; *Elkhadragy et al., 2018*). Similarly, IL-8 has been reported to enhance migration and thus metastasis of breast carcinoma cells MDA-MB231, which also carry a KRAS G13D mutation (*Jayatilaka et al., 2017*). Considering that ERK3 regulates IL-8 production in many tested cell lines, including MDA-MB231, we investigated the metastatic potential of control (shCo) and ERK3-depleted (shERK3) MDA-MB231 cells (*Figure 5C–F*). Intravenous (i.v) injection of ERK3-depleted (shERK3) MDA-MB231 cells into the tail vein of immunocompromised mice resulted in less tumor lesions in the lungs and decreased pulmonary metastatic burden (*Figure 5E and F*, respectively and *Figure 5—figure supplement 1*). These results demonstrate a critical role of ERK3 in mediating breast cancer cell seeding and lung metastasis.

## ERK3 positively regulates activity of IL-8 promoter and DNA binding activity of AP-1

Since ERK3 depletion led to a significant decrease in *CXCL8* mRNA expression, we tested whether ERK3 controls promoter activity of IL-8. Gaussia Luciferase (GLuc) promoter reporter assay revealed that the depletion of ERK3 with siRNAs reduced IL-8 promoter activity and consistently IL-8 production in the same cells (*Figure 6A–C*). To determine which transcription factors are regulated by ERK3 and therefore are responsible for the observed decrease in IL-8 promoter activity in ERK3-depleted cells, we performed a transcription factors (TFs) activity profiling array. As presented in *Figure 6—figure supplement 1*, ERK3 is required for the activation of the majority of the tested TFs in HCPECs, including AP-1 (*Figure 6D*) which has been reported to control both, basal and inducible expression of chemokines like IL-8 (*Khanjani et al., 2012*; *O'Hara et al., 2009*). These results were further independently validated by employing specific TF activity assays measuring DNA binding activity of three most-potent regulators of IL-8, which apart from AP-1/c-Jun, also involves C/EBP and CREB (*Hoffmann et al., 2002*; *Jundi and Greene, 2015*). Our results highlighted AP-1 as one of the TFs positively regulated by ERK3 in HCPECs (*Figure 6E* and *Figure 6—figure supplement 2*). Interestingly, in contrast to AP-1, NF-κB activity was increased in the ERK3-depleted HCPECs (*Figure 6—figure supplement 3*), further implicating AP-1 as the transcription factor primarily contributing to IL-8 in response to ERK3 depletion under these settings (*Figure 6E–F*).

Considering that ERK3 regulates basal activity of AP-1 and thus expression levels of IL-8 in HCPECs, we further tested whether ERK3 is also required for LPS-induced AP-1 activity in HT-29 cells. AP-1 TF filter plate assay revealed that LPS-induced AP-1 DNA binding activity and IL-8 production are dependent on ERK3 protein levels (*Figure 6G–H*).

## ERK3-dependent regulation of AP-1 signaling: LPS-induced nuclear co-localization of ERK3 and c-Jun

In order to uncover the mechanisms underlying the function of AP-1 within the ERK3-IL-8 axis, we tested whether ERK3 forms a complex with c-Jun in HT-29 cells. We detected a complex formation between these two proteins upon LPS stimulation in cells by performing two-way

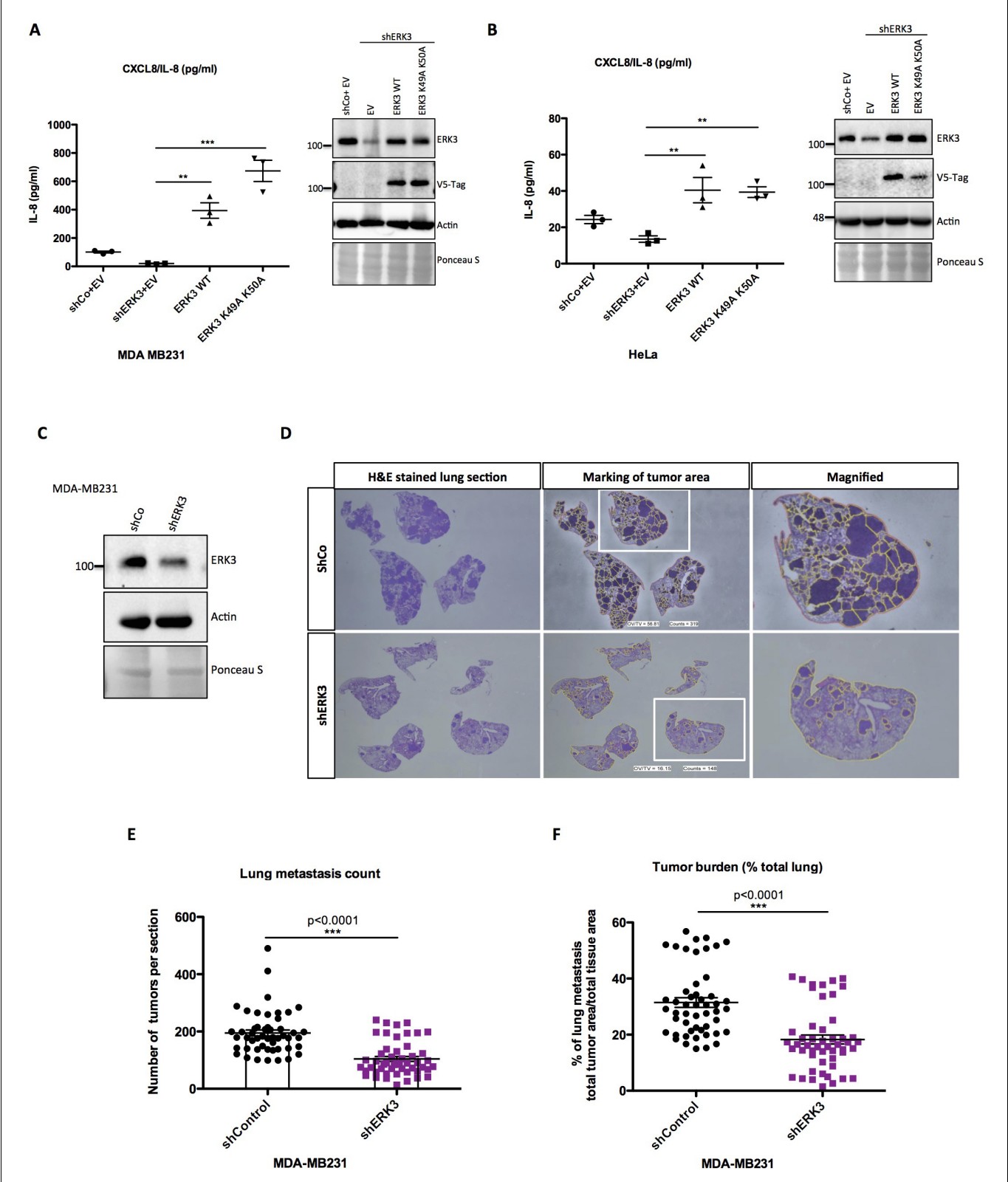

**Figure 5.** ERK3 regulates levels of IL-8 in a kinase-independent manner and promotes lung metastasis of MDA-MB231 breast cancer cells. (A–B) ERK3 regulates IL-8 production in a kinase-independent manner. (A) MDA-MB231 ERK3 knockdown (shERK3) cells were reconstituted with wild type (ERK3 WT), ERK3 kinase dead mutant (ERK3 K49A K50A) or with an empty vector control (EV). IL-8 levels were determined in obtained supernatants by ELISA. IL-8 concentrations are presented as mean ± SEM in pg/ml from three biological replicates (n = 3); *p<0.05, **p<0.01, ***p<0.001, one-way ANOVA,

*Figure 5 continued on next page*

*Figure 5 continued*

Turkey's post-test. Immunoblot analysis was performed and levels of ERK3 protein were monitored to determine ERK3 knockdown efficiency and the expression of exogenously expressed constructs (V5-Tag expression). Actin and Ponceau S staining were used as loading controls. (B) HeLa control and ERK3-depleted cells were transiently transfected with wild type ERK3, kinase dead mutant K49A K50A and empty vector control. IL-8 levels were measured as described previously and are represented as mean ± SEM concentrations from three biological replicates (n = 3); *p<0.05, **p<0.01, ***p<0.001, one-way ANOVA, Turkey's post-test. Knockdown and overexpression efficiency were verified by immunoblot analysis. Actin and Ponceau S staining were used as loading controls. (C-D) ERK3 knockdown leads to a decrease in lung metastasis of MDA-MB231 breast cancer cells. Tail-vein injection of control (shCo) or ERK3-depleted (shERK3) MDA-MB231 cells was performed (n = 5). (C) Western blot analysis of MDA-MB231 cells stably transfected with either shRNA targeting ERK3 (shERK3) or control shRNA (shCo). Levels of ERK3 were determined to estimate knockdown efficiency, actin and Ponceau S are provided as loading controls. (D) Representative images of H and E stained lungs sections photographed with Nikon D90 digital camera (H and E stained lung section), marking of the tumor areas performed in ImageJ (marking of the tumor area), boxed areas were magnified and are shown on the right (magnified). (E-F) ImageJ quantification was performed using color deconvolution of five sections per lung, representing: (E) the exact numbers of lung tumors per each analyzed section. (F) Pulmonary metastatic burden expressed as percentage of the tumor in evaluated lung tissue (total tumor area/total tissue area); *p<0.05, **p<0.01, ***p<0.001, t-test. Please find *Figure 5—figure supplement 1* for graphs representing mean ± SEM per each mouse (n = 5), for both, tumor lesions number and tumor burden.
The online version of this article includes the following source data and figure supplement(s) for figure 5:

**Source data 1.** Full membrane scans for western blot images for *Figure 5A–C*.
**Figure supplement 1.** ImageJ quantification of tumor lesions using color deconvolution.

immunoprecipitation (IP) assays with validated antibodies directed against both c-Jun and ERK3 antigens (*Figure 7A*).

Cell fractionation and immunofluorescence (IF) studies confirmed that while total ERK3 and c-Jun protein levels increase upon LPS stimulation, a significant fraction of ERK3 can be detected in the nuclear compartments, ultimately leading to an enhanced nuclear abundance of c-Jun protein upon stimulation (*Figure 7B–D* and *Figure 7—figure supplement 1*). Further, single-cell level analyses revealed a significant nuclear co-localization between ERK3 and c-Jun proteins in both control and LPS stimulated HT-29 cells with Pearson's correlation coefficient PCC: 0.9172 ± 0.0098 and PCC: 0.9237 ± 0.01 for control and LPS stimulated cells, respectively (mean ± SEM) (*Figure 7E*). Knockdown of ERK3 expectedly abolished the observed nuclear co-localization (*Figure 7B and E*) and interestingly further reduced LPS-induced nuclear abundance of c-Jun (*Figure 7B and F*). These data suggested a critical role for ERK3 in maintaining the nuclear abundance of c-Jun and thus AP-1 activity.

## ERK3 maintains transcriptional regulation of epithelial secretome and IL-8 mediated human leukocyte chemotaxis in vitro and in vivo

We next explored the physiological significance of the ERK3-IL-8 signaling pathway by performing chemotaxis experiments in vitro and in vivo, as described in the Materials and methods. Transwell migration experiments involving THP1 cells and human neutrophils revealed that depletion of ERK3 in the epithelial cells leads to a significant reduction in the chemotaxis of these leukocytes (*Figure 8A–B*). We then performed similar experiments in vivo by injecting the secretome of control or ERK3-depleted HCPECs into the peritoneal cavity of mice as presented in the experimental scheme (*Figure 8C*). Consistent with the observations we made in vitro, injection of the supernatants obtained from ERK3-depleted HCPECs (siERK3 supernatant) resulted in significantly reduced migration of granulocytes and monocytes to the intraperitoneal cavity in comparison to the controls (*Figure 8D–E*), while not exerting any significant effect on the lymphocytes recruitment (*Figure 8F*). To further reaffirm the role of IL-8 in mediating the chemotaxis of leukocytes, we employed an IL-8 neutralizing antibody. As shown in *Figure 8A and B*, the presence of the IL-8 antibody decreased the migration of neutrophils and THP1 cells. Moreover, neutralization antibody incubation indicated that the observed decrease in the chemotaxis is mediated by IL-8, as there is an additive effect between untreated and antibody treated variant within each condition (Control-Control+IL-8 Ab and ERK3 KD-ERK3 KD+IL-8 Ab). Importantly, although mice lack the IL-8 encoding gene, they do express a receptor analogous to human CXCR2, which in response to human IL-8 mediates neutrophil chemotaxis (*Singer and Sansonetti, 2004*). As expected, intraperitoneal injection of human recombinant IL-8 into mice significantly attracted leukocytes confirming the crucial role for IL-8 in this process (*Figure 8D–F*).

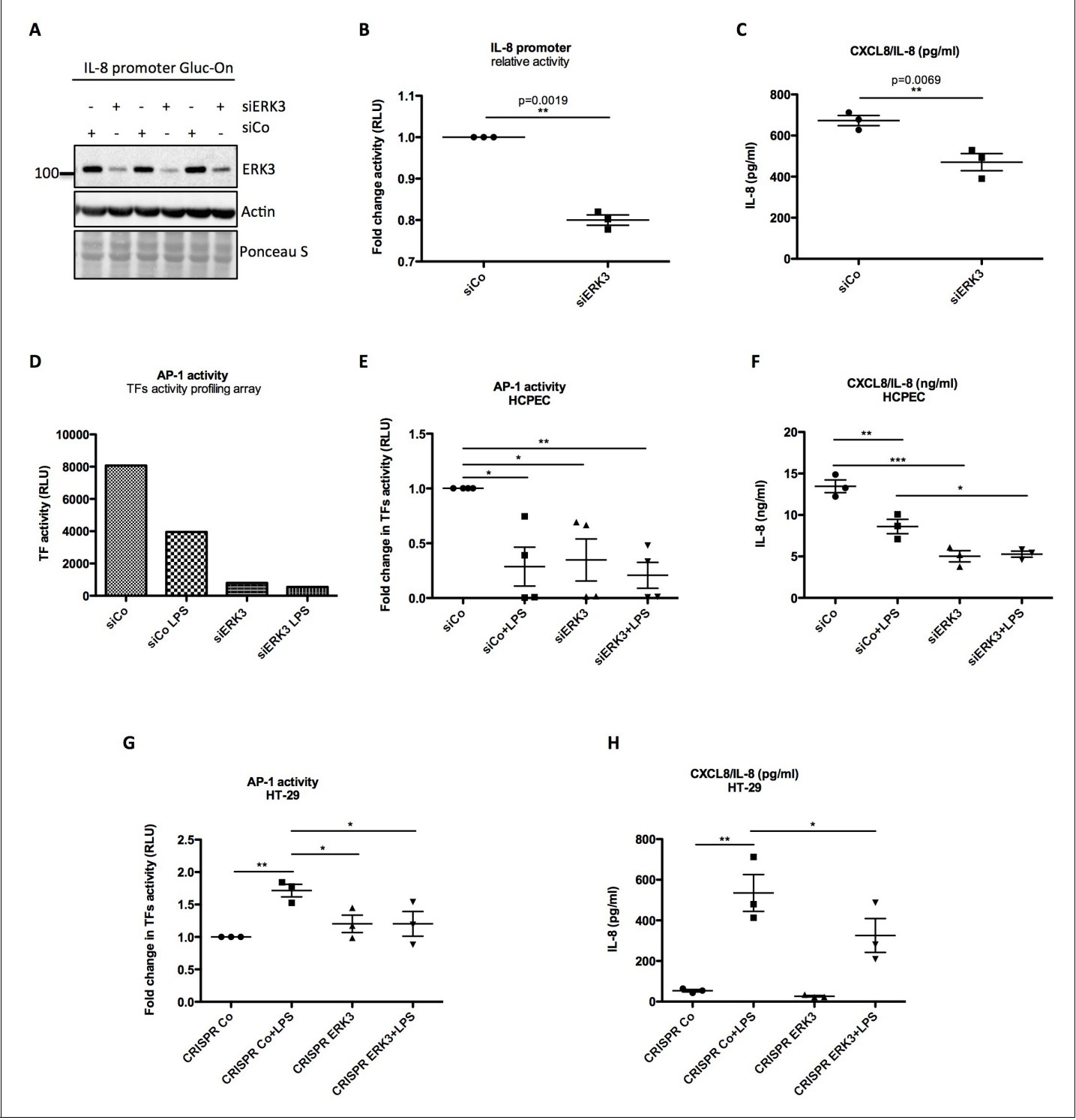

**Figure 6.** ERK3 positively regulates activity of IL-8 promoter and DNA binding activity of AP-1. (**A–C**) IL-8 promoter activity is decreased in ERK3-depleted cells. MDA-MB231 cells were stably transfected with CXCL8 Gaussia Luciferase construct (Gluc-On Promoter Reporter Clones). Cells were further transfected with siRNA targeting ERK3 (siERK3) or control siRNA (siCo). 24 hr post-transfection, medium was exchanged for DMEM-FBS, cells were cultured for additional 24 hr and supernatants were harvested. (**A**) Cells were subjected to SDS-PAGE analysis. Knockdown efficiency of ERK3 was monitored. Actin and Ponceau S staining were performed as loading controls. (**B**) Luciferase activity was monitored and shown here is mean ± SEM fold change in RLU of siERK3 samples normalized to the siCo (n = 3); *p<0.05, **p<0.01, ***p<0.001, paired t-test. (**C**) IL-8 levels were measured by ELISA and mean ± SEM concentrations in pg/ml are presented from three biological replicates (n = 3); *p<0.05, **p<0.01, ***p<0.001, t-test. (**D**) TFs activity profiling assay. HCPECs were transfected with either a negative control siRNA (siCo) or siRNA targeting ERK3. 24 hr post-transfection medium was

*Figure 6 continued on next page*

Figure 6 continued

changed for MEM minus FBS and other supplements and cells were stimulated with LPS for 24 hr. After stimulation, cell culture supernatants were harvested from each dish and part of the cells was lysed in RIPA buffer for further western blot analysis and knockdown verification presented in *Figure 6—figure supplement 3B*. The remaining cells were subjected to nuclear extraction and further TFs activation plate profiling (*Figure 6—figure supplement 1* and *Figure 6—source data 1*). Graph presented in (D) depicts transcriptional activity of AP-1 as fold change in RLU measured by TFs activity profiling array as mentioned in the Materials and methods section. (E) Graphical representation of AP-1 activity measured with filter plate assay according to the manufacturer's instructions. 24 hr post-transfection, control (siCo) and ERK3 knockdown (siERK3) HCPECs were stimulated with LPS for 24 hr in medium without any supplements. Afterwards, cells were subjected to either western blot analysis or nuclear extraction and filter plate assay analysis. Results are represented as mean fold change in activity measured in RLU ± SEM from four independent experiments (n = 4); *p<0.05, **p<0.01, ***p<0.001, one-way ANOVA, Turkey's post-test. (F) ELISA of IL-8 levels measured in control and LPS stimulated siCo/siERK3 HCPECs. Results are depicted as mean concentration (pg/ml)± SEM from three biological replicates (n = 3) per condition; *p<0.05, **p<0.01, ***p<0.001, one-way ANOVA, Turkey's post-test. Activity of two other TFs involved in the regulation of IL-8 was assessed in control and ERK3-depleted HCPECs and graphs are presented in *Figure 6—figure supplement 2*. (G-H) LPS-induced AP-1 activity in HT-29 cells is impaired by ERK3 knockout (CRISPR ERK3) which leads to a decrease in IL-8 levels (G) Graph represents AP-1 binding activity analysis by filter plate assay in control (CRISPR Co) and ERK3 knockout (CRISPR ERK3) cells in the presence and absence of LPS. Data are presented as mean fold change in RLU ± SEM from three independent experiments (n = 3); *p<0.05, **p<0.01, ***p<0.001, one-way ANOVA, Turkey's post-test. (H) IL-8 concentration in pg/ml is presented as mean ± SEM obtained from three independent experiments (n = 3); *p<0.05, **p<0.01, ***p<0.001, one-way ANOVA, Turkey's post-test.

The online version of this article includes the following source data and figure supplement(s) for figure 6:

**Source data 1.** Full membrane scans for western blot images for *Figure 6A*.
**Source data 2.** Transcription factors (TFs) activity profiling array.
**Figure supplement 1.** TFs activity profiling array.
**Figure supplement 2.** Graphical representation of (A) C/EBP and (B) CREB activity measured in control (siCo) and ERK3-depleted (siERK3) HCPECs with filter plate assay according to the manufacturer's instructions.
**Figure supplement 3.** Activation of NFkB in control and ERK3 knock down cells upon LPS stimulation.
**Figure supplement 3—source data 1.** Full membrane scans for western blot images for *Figure 6—figure supplement 3B*.

Previous studies showed that IL-8 upregulates intercellular adhesion molecule-1 (ICAM-1) in HT-29 cells, thus promoting neutrophil-epithelium cell adhesion (*Kelly et al., 1994*). Moreover, studies on epithelial cells revealed that secreted IL-8 enhances the expression of CD11/CD18 adhesion receptors by neutrophils, which enables the migration by promoting the interaction of the receptors with ICAM-1 (*Kelly et al., 1994*). Interestingly, in addition to IL-8, ERK3 also positively regulates *ICAM-1* mRNA expression as revealed by the RNAseq analysis (*Figure 8—figure supplement 1*).

These results unveil ERK3 as a novel interacting partner of c-Jun and regulator of AP-1 activity and confirm the crucial role for the ERK3-IL-8 signaling axis in mediating epithelial chemotaxis both in vitro and in vivo (*Figure 9*).

## Discussion

MAPKs are members of a highly conserved kinase family, and the role of classical MAPKs in regulating innate immune responses is well documented (*Arthur and Ley, 2013*; *Marie et al., 1999*; *Newton and Dixit, 2012*). The intestinal epithelium serves as the first line of defence against pathogen entry and bacteria-derived products like LPS trigger TLR signaling, leading to the production of cytokines and chemokines that attract immune cells to the site of infection, thus initiating local inflammation (*Abreu et al., 2002*; *Cario and Podolsky, 2000*; *Kawai and Akira, 2010*; *Mogensen, 2009*; *Sallusto and Baggiolini, 2008*; *Singer and Sansonetti, 2004*). While the classical MAPKs have been shown to be activated and required for innate immune responses, the role of atypical MAPKs remains understudied.

This study demonstrates a role for ERK3 in regulating epithelial secretome and LPS-mediated immune responses. Although ERK3 protein levels alter during epithelial differentiation (*Figure 1*), loss-of-function studies revealed that ERK3 is probably not required for the establishment and maintenance of human gastric epithelium (*Figure 4I–J*). However, incorporation of the HGOs confirmed that depletion of ERK3 causes a decrease in CXCL8/IL-8 secretion by epithelial cells (*Figure 4K*). We observed that LPS treatment affected ERK3 protein stability in a cell-type-dependent manner. While LPS led to an enhanced ERK3 protein stability in oncogenic cells, opposing effects were observed in human primary colonic epithelial cells (*Figure 2*). Unlike the tumorigenic HT-29 cells, primary cells express low levels of *TLR4* (*Figure 2—figure supplement 2*). Whether this discrepancy contributes

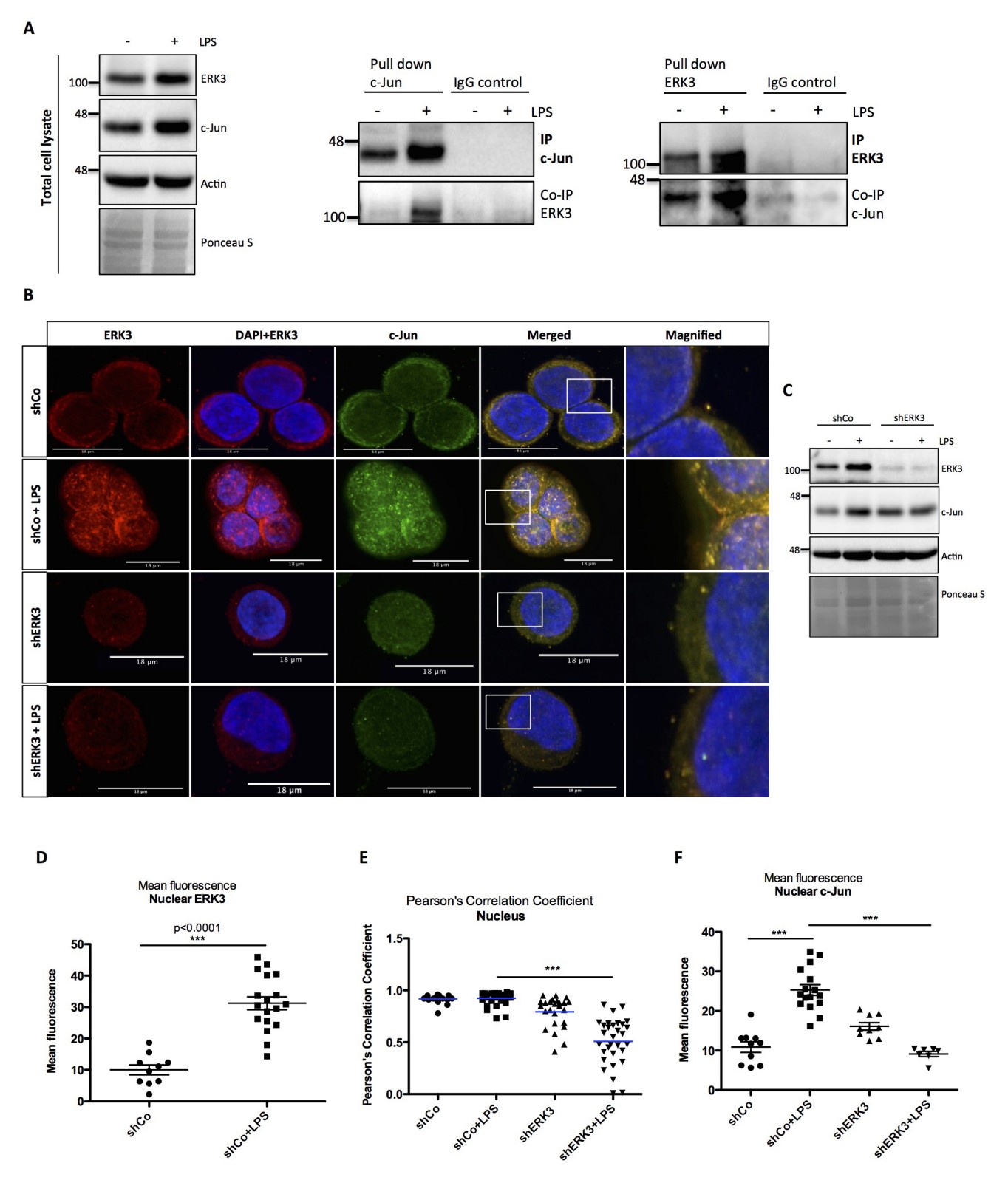

**Figure 7.** ERK3 interacts with c-Jun and regulates its nuclear abundance. (**A**) Co-immunoprecipitation (IP) of ERK3 and c-Jun in unstimulated and LPS stimulated HT-29 cells using a c-Jun or ERK3 antibody. Levels of c-Jun and ERK3 were monitored. IgG isotype control for IP and co-IP was included. Actin and Ponceau S staining were used as loading controls for total cell lysate western blot analysis. (**B**) Confocal analysis of IF staining of control (shCo) and ERK3 knockdown (shERK3) HT-29 cells cultured in the presence and absence of LPS. Cells were stained with c-Jun primary antibody followed

*Figure 7 continued on next page*

*Figure 7 continued*

by rabbit Alexa488 (green), with ERK3 antibody followed by Cy3 mouse secondary (red) and Hoechst for the nuclei. Scale bars 18 μm. Boxed areas in merged images were magnified and are presented on the right. (C) Western blot analysis of control (shCo) and ERK3-depleted (shERK) HT-29 cells subjected to IF staining presented in C. Levels of ERK3 and c-Jun are depicted in the control and LPS stimulated cells as well as actin loading control and Ponceau S staining. (D) ImageJ quantification of the fluorescence intensities was performed as described in the Materials and methods section. Graph represents mean red (ERK3) fluorescence intensities in control (shCo) and LPS stimulated (shCo+LPS) HT-29 cells; *p<0.05, **p<0.01, ***p<0.001, one-way ANOVA, Turkey's post-test. (E) Pearson's correlation coefficient values obtained from co-localization analyses as described in the Materials and methods section are presented in control (shCo) and ERK3 knockdown cells (shERK) under -/+ LPS conditions. Scores above 0 indicate a tendency towards co-localization with a perfect co-localization with a score of 1; *p<0.05, **p<0.01, ***p<0.001, one-way ANOVA, Turkey's post-test. (F) Mean green fluorescence intensity (c-Jun) was determined in control (shCo) and ERK3-depleted (shERK) HT-29 cells in the presence and absence of LPS; *p<0.05, **p<0.01, ***p<0.001, one-way ANOVA, Turkey's post-test.

The online version of this article includes the following source data and figure supplement(s) for figure 7:

**Source data 1.** Full membrane scans for western blot images for *Figure 7A and C*.

**Figure supplement 1.** Cell fractionation experiment performed in HT-29 cells in the presence and absence of LPS (1.5 hr).

**Figure supplement 1—source data 1.** Full membrane scans for western blot images for *Figure 7—figure supplement 1*.

---

to the effects observed on ERK3 protein levels remains unclear (*Cario and Podolsky, 2000*; *Tang et al., 2010*). Interestingly, upregulation of TLR4 is associated with inflammatory bowel diseases like ulcerative colitis (*Cario and Podolsky, 2000*; *Fan and Liu, 2015*; *Tang et al., 2010*). Moreover, observed discrepancies in responsiveness to LPS between colon carcinoma and primary epithelial cells can be further explained by previous studies describing upregulation of TLR4 in colorectal cancer, including HT-29 cells (*Abreu et al., 2002*; *Furrie et al., 2005*). Published observations indicate that LPS-TLR4 signaling results in tumor progression and metastasis (*Gross et al., 1995*; *Tang et al., 2010*; *Yesudhas et al., 2014*). Intestinal epithelium is constantly exposed to significant levels of luminal LPS and the low expression of TLR4 by HCPECs might provide an explanation why they can tolerate LPS and maintain hyporesponsiveness toward luminal bacteria (*Otte et al., 2004*). It is tempting to hypothesize that LPS triggers alterations in the ubiquitin signaling machinery contributing to ERK3 ubiquitination and degradation, leading to the attenuation of IL-8 production and inflammation in primary epithelium. While Usp20 has been shown to function as a DUB for ERK3, the E3 ubiquitin ligase of ERK3 is not known (*Mathien et al., 2017*). Also, the kind of ubiquitin chains synthesized on ERK3 in response to LPS deserves further investigations.

In order to gain more insight into the immune stimuli-dependent ERK3 kinetics in intestine epithelial cells, we tested two other TLR ligands, Pam3CSK4 (for TLR1/2) and R848 (for TLR7/8) as well as IL-1β. These experiments revealed that ERK3 is required in a stimulus- and cell-type-dependent manner for the production of IL-8. Further studies are warranted to uncover the underlying mechanisms contributing to these discrepancies. However, ERK3 is critically required for basal as well as LPS-induced IL-8 levels in both primary and tumor cells. IL-8 was first discovered as a leukocyte chemotactic factor and since then emerged as a double-edged sword of inflammation (*Beck et al., 2016*; *Hammond et al., 1995*; *Long et al., 2016*; *Sallusto and Baggiolini, 2008*; *Singer and Sansonetti, 2004*). Further, IL-8 has emerged as a crucial factor in mediating tumor angiogenesis, tumor cell survival and metastasis (*David et al., 2016*; *Feng et al., 2018*; *Itoh et al., 2005*; *Li et al., 2003*; *Waugh and Wilson, 2008*; *Xu and Fidler, 2001*). Loss-of-function studies in multiple cell lines confirmed the obligatory role for ERK3 in controlling IL-8 levels and several other chemokines and cytokines including *CCL2*, *CXCL6* or adhesion molecules like *ICAM-1* in HCPECs. Consequently, CRISPR/Cas9 knockout of ERK3 negatively affected secretion of CXCL8/IL-8 in human gastric organoids. Depletion of ERK3 from MDA-MB231 cells significantly inhibited breast cancer lung metastasis burden in mice (*Figure 5C–F* and *Figure 5—figure supplement 1*). These data suggest a crucial role for ERK3 in IL-8-mediated tumor progression and metastasis formation by regulating critical factors like CXCL8/IL-8.

Intriguingly, the effect is kinase-independent and we demonstrated that ERK3 controls the transcription factor AP-1/c-Jun in both, HCPECs and HT-29 cells (*Figure 6D–E* and *Figure 6G*, respectively). We further observed that ERK3-AP-1 activity directly correlates with IL-8 levels in both normal and transformed cells (*Figure 6E–F and G–H*, respectively).

In order to activate target genes, transcription factors such as c-Jun have to translocate to the nucleus (*Schreck et al., 2011*). Our studies revealed that ERK3 is required for nuclear abundance of

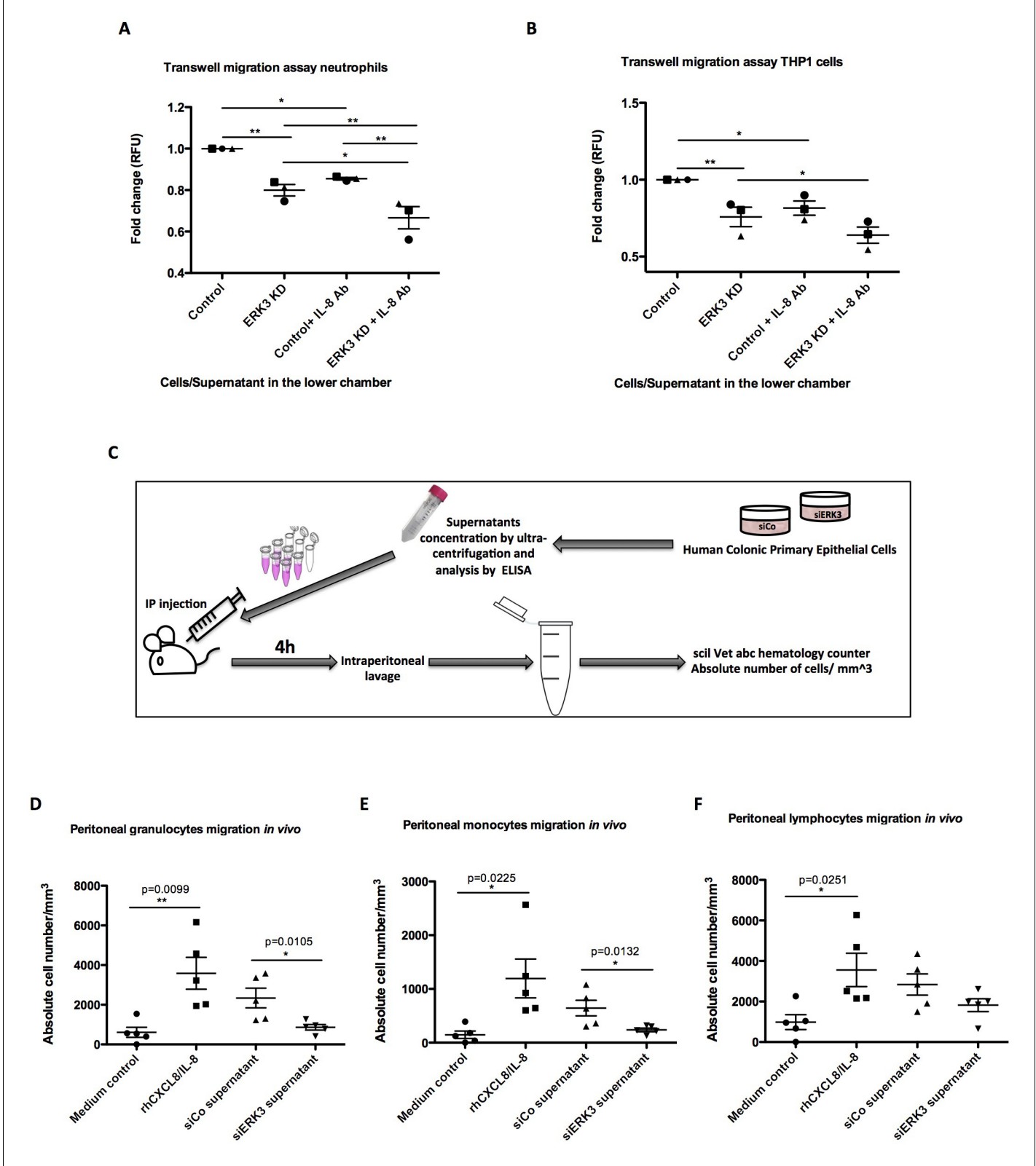

**Figure 8.** ERK3 regulates epithelial secretome and IL-8-mediated chemotaxis of human neutrophils and monocytes in vitro and in vivo. (A–B) In vitro migration of (A) neutrophils and (B) THP1 cells toward epithelium. CellTracker Green-stained cells were placed in the upper chamber of the transwell plate and supernatants obtained from control and ERK3-depleted HCPECs or HT-29 cells were placed in the lower chamber. IL-8 neutralizing antibody was used in each condition as a control. Following 2 hr incubation at 37°C, migration of neutrophils or THP1 cells to the lower chamber was measured
*Figure 8 continued on next page*

*Figure 8 continued*

using fluorescence. Fold change of Relative Fluorescence Units (RFU) was then calculated. Data represent mean ± SEM of three biological replicates (n = 3); *p<0.05, **p<0.01, ***p<0.001, one-way ANOVA, Turkey's post-test. (**C–F**) Effect of ERK3-depleted epithelial supernatants on intraperitoneal leukocytes migration in vivo. The experimental procedure is explained in detail in Materials and methods. Briefly, groups of five 8-week-old C57BL/6J female mice were injected intraperitoneally (i.p) with one of the following: MEM without supplements (MEM control), MEM containing 900 ng of human recombinant CXCL8/IL-8 (rhCXCL8/IL-8), HCPEC siCo/siERK3-derived concentrated supernatants. 4 hr post-injections mice were sacrificed and peritoneal white blood cells populations were harvested by peritoneal lavage. (**D–F**) Scatter plots representing absolute number of (**D**) granulocytes, (**E**) monocytes and (**F**) lymphocytes. Data are represented as mean ± SEM, n = 5; *p<0.05, **p<0.01, ***p<0.001, t-test.

The online version of this article includes the following figure supplement(s) for figure 8:

**Figure supplement 1.** ERK3 regulates *ICAM-1* expression.

c-Jun protein (***Figure 7B and F***). We provide evidence that ERK3 interacts with c-Jun and regulates the DNA-binding activity of AP-1 (***Figure 6D–E and G*** and ***Figure 7***) which might reflect a non-catalytic scaffolding function described for other kinases as well (***Rauch et al., 2011***). Activated JNK translocates to the nucleus, where it can phosphorylate c-Jun at S63/73 and T91/93, which in turn

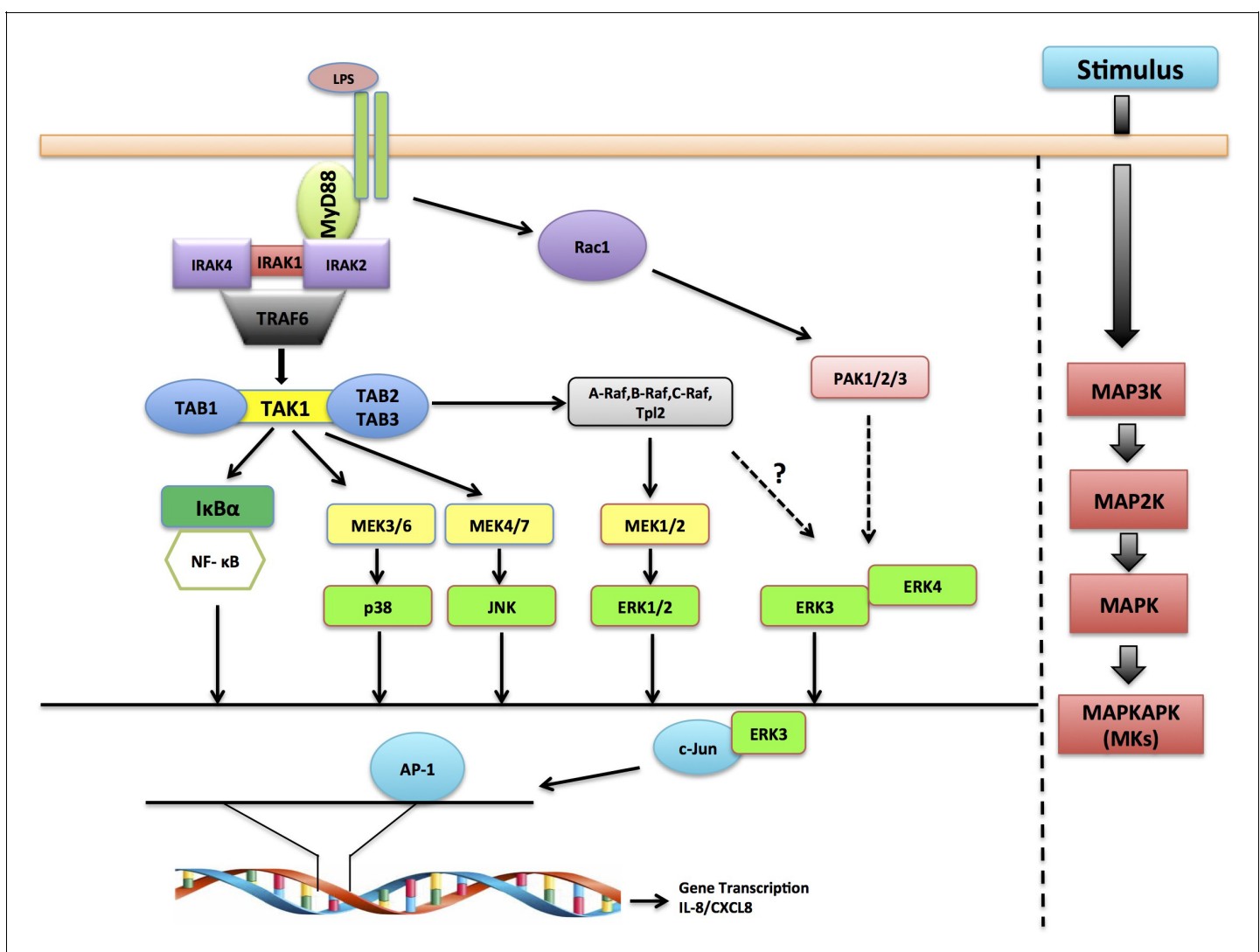

**Figure 9.** Schematic representation depicting the role of ERK3 in chemotaxis. ERK3 interacts with c-Jun and by regulating its nuclear abundance controls DNA-binding activity of AP-1 TF, which is critically required for the activation of several cytokines, including IL-8 that contribute to the chemotaxis of leukocytes to the epithelium in response to LPS and other innate immune stimuli.

enables c-Jun homodimerization or heterodimerization with c-Fos (*Deng and Karin, 1994*; *Gazon et al., 2017*). Nevertheless, JNK controls c-Jun activity by sub-nuclear localization of AP-1 proteins (*Gazon et al., 2017*), therefore it is tempting to propose ERK3 as a novel MAPK required for the nuclear abundance of c-Jun. Whether ERK3 directly interacts with c-Jun and influences the stoichiometry of the AP-1 complex needs further studies.

ERK3 has already been reported to be constitutively localized to both, cytoplasmic and nuclear compartments (*Julien et al., 2003*), but its subcellular localization has been shown not to be affected by any of the common MAPKs activating stimuli (*Julien et al., 2003*). Here, we demonstrate that LPS stimulation triggers the accumulation of ERK3 protein, resulting in its enhanced expression in both, cytoplasm and nucleus (*Figure 7B–D* and *Figure 7—figure supplement 1*).

Furthermore, we consistently observed that loss of ERK3 inhibited IL-8 production and secretion in multiple cell types - despite the activation of classical MAPKs and NF-κB. Interestingly, NF-κB activity was increased upon ERK3 knockdown as indicated by TF activity profiling assay as well as by measuring IκBα degradation (*Figure 6—figure supplement 3A and B*, respectively), which is contradictory to the observed decrease in IL-8 production (*Figure 6F*). Worth mentioning is that LPS stimulation led to a downregulation of NF-κB activity in ERK3-depleted cells (*Figure 6—figure supplement 3A*), which surprisingly had no additive effect on neither protein (*Figure 6F*) nor mRNA (*Figure 4—figure supplement 1A*) levels of CXCL8/IL-8. The possible cross-talk between ERK3 and NF-κB deserves further investigations.

We were able to observe that inhibition of MEK1 with trametinib reduces the protein stability of ERK3 and thus the published results claiming a unique role for the classical MAPK in the production of IL-8 needs to be evaluated with caution. Our data suggest an obligatory and synergistic role for ERK3 in mediating LPS-induced chemokines. We also evaluated the role of ERK4 and MK5 in the regulation of ERK3 and IL-8. It is not clear if the heteromerization between these kinases is contributing to their protein stability.

While verifying the role of ERK3 in the regulation of CXCL8/IL-8, we uncover that ERK3 interacts with c-Jun and regulates AP-1 activity, which directly contributes to the high basal and LPS-mediated IL-8 levels in HCPECs and HT-29 cells, respectively. Although HCPECs and HT-29 cells respond differently to LPS, mechanisms regulating IL-8 in both cell types are ERK3-AP-1 dependent and the decrease in expression of ERK3 directly correlates with IL-8 levels. Taken together, these data present one of the first physiological roles for this understudied MAPK and unveil a critical role for ERK3-IL-8 signaling axis in regulating epithelial function and chemotaxis, which is critical for mediating inflammation and tumorigenesis.

# Materials and methods

## Key resources table

| Reagent type (species) or resource | Designation | Source or reference | Identifiers | Additional information |
|---|---|---|---|---|
| Cell line (*Homo sapiens*) | HT-29 | ATCC | ATCC HTB-38 | |
| Cell line (*Homo sapiens*) | Human Colonic Primary Epithelial Cells (HCPECs) | ATCC | CCD 841 CoN, ATCC–CRL-1790 | |
| Cell line (*Homo sapiens*) | CaCo2 | kind gift from Prof. Monilola Olayioye (University of Stuttgart) | | |
| Cell line (*Homo sapiens*) | MDA-MB231 | DSMZ | ACC 732 | |
| Cell line (*Homo sapiens*) | THP1 | DSMZ | ACC 16 | |
| Cell line (*Homo sapiens*) | 293 T cells | kind gift from Dr. Andreas Ernst (Goethe-University Frankfurt am Main, IBC2) | | |

*Continued on next page*

*Continued*

| Reagent type (species) or resource | Designation | Source or reference | Identifiers | Additional information |
|---|---|---|---|---|
| Chemical compound, drug | LPS | Sigma | Cat# L6143 | Working concentration: 200 ng/ml |
| Chemical compound, drug | Human recombinant IL-1β | ImmunoTools | Cat# 11340013 | Working concentration: 10 ng/ml |
| Chemical compound, drug | Pam3CSK4 TLR1/2 ligand | InvivoGen | Cat# tlrl-pms | Working concentration: 20 µg/ml |
| Chemical compound, drug | R848 TLR7/8 ligand | InvivoGen | Cat# tlrl-r848 | Working concentration: 2.5 µg/ml |
| Chemical compound, drug | Cycloheximide (CHX) | Sigma | Cat# C-7698 | Working concentration: 100 µg/ml |
| Chemical compound, drug | MG-132 inhibitor | Calbiochem, Merck Millipore | Cat# 474790 | Working concentration: 10 µM |
| Chemical compound, drug | Selective MEK1/2 inhibitor trametinib GSK1120212 | Selleckchem | Cat# S2673 | Working concentration: 1 µM |
| Sequence-based reagent-shRNA | shMAPK6#1 (shERK3#1) NM_002748.x-3734s1c1 | MISSION shRNA Human Library (Sigma) | TRCN0000001568 | Sequence: CCGGGCTGTCCACGTAC TTAATTTACTCGAGTAAATT AAGTACGTGGACAGCTTTTT |
| Sequence-based reagent-shRNA | shMAPK6#2 (shERK3#2) NM_002748.x-1564s1c1 | MISSION shRNA Human Library (Sigma) | TRCN0000001569 | Sequence: CCGGGACATGACTGAGCCA CACAAACTCGAGTTTGTGT GGCTCAGTCATGTCTTTTT |
| Sequence-based reagent-shRNA | shMAPK6#3 (shERK3#3) NM_002748.x-798s1c1 | MISSION shRNA Human Library (Sigma) | TRCN0000001570 | Sequence: CCGGTGATCTGGGTTCT AGGTATATCTCGAGATATAC CTAGAACCCAGATCATTTTT |
| Sequence-based reagent-shRNA | shMAPKAPK5#1 (shMK5#1) NM_003668.2–475 s1c1 | MISSION shRNA Human Library (Sigma) | TRCN0000194823 | Sequence: CCGGCCCAAACATAGTTCA GATTATCTCGAGATAATCTGA ACTATGTTTGGGTTTTTTG |
| Sequence-based reagent-shRNA | shMAPKAPK5#5 (shMK5#5) NM_003668.x-1622s1c1 | MISSION shRNA Human Library (Sigma) | TRCN0000000682 | Sequence: CCGGGAAATTGTGAAGCA GGTGATACTCGAGTATCACC TGCTTCACAATTTCTTTTT |
| Sequence-based reagent-shRNA | shMAPK4#1 (shERK4#1) NM_002747.x-3808s1c1 | MISSION shRNA Human Library (Sigma) | TRCN0000001374 | Sequence: CCGGCTCACACCACACGCCT TAAATCTCGAGATTTAAGGC GTGTGGTGTGAGTTTTT |
| Sequence-based reagent-shRNA | shMAPK4#2 (shERK4#1) NM_002747.x-1105s1c1 | MISSION shRNA Human Library (Sigma) | TRCN0000001375 | Sequence: CCGGACTACACCAAAGCCAT CGACACTCGAGTGTCGATGG CTTTGGTGTAGTTTTTT |
| Sequence-based reagent-shRNA | shMAPK4#5 (shERK4#5) NM_002747.x-1017s1c1 | MISSION shRNA Human Library (Sigma) | TRCN0000001378 | Sequence: CCGGGGATCAGCATTACTCCC ACAAGCTCGAGCTTGTGGGA GTAATGCTGATCTTTTT |
| Sequence-based reagent-siRNA | siMAPK6#1 (siERK3#1) | FlexiTube siRNA, Hs_MAPK6_5 Qiagen | Cat# SI00606025 | Sequence: AGUUCAAUUUGAAAGGAAATT |
| Sequence-based reagent-CRISPR/Cas9 | CRISPR ERK3 | designed by Rule Set 2 of Azimuth 2.0 as described previously (*Doench et al., 2016*) | | Selected gRNAs sequence: #1 5'-CACCGGAGCCAATTAACAG ACGATGT-3' #2 5'-CACCGATACTTGTAACTA CAAAACG-3' #3 5'-CACCGCTGCTGTTAACC GATCCATG-3' |

*Continued on next page*

*Continued*

| Reagent type (species) or resource | Designation | Source or reference | Identifiers | Additional information |
|---|---|---|---|---|
| Recombinant DNA reagent-cDNA | ERK3 K49A K50A kinase dead mutant | Site-directed mutagenesis | | Primers sequence: frw_5' GCAATTGTCCTTACT GATCCCCAGAGTGTC, rev_5' CGCGATGGCTACTC TTTTGTCACAGTC |
| Commercial assay, kit | Human IL-8 ELISA Ready-SET-Go! Kit | eBioscience | Cat# 88–8086 | |
| Commercial assay, kit | Cell Fractionation Kit | Invent Biotechnologies | Cat# SM-005 | |
| Commercial assay, kit | RayBiotech L-Series Human Antibody Array 1000 | Tebu-Bio | Cat# AAH-BLG-1000 | |
| Commercial assay, kit | Nuclear Extraction Kit | Signosis | Cat# SK-0001 | |
| Commercial assay, kit | TF Activation Profiling Plate Array I | Signosis | Cat# FA-1001 | |
| Commercial assay, kit | Transcription factor Filter Plate Assay AP-1 probe | Signosis | Cat# FA-0004 | |
| Commercial assay, kit | Transcription factor Filter Plate Assay C/EBP probe | Signosis | Cat# PP-0011 | |
| Commercial assay, kit | Transcription factor Filter Plate Assay CREB probe | Signosis | Cat# PP-0015 | |
| Commercial assay, kit | CXCL8-Gaussia Luciferase GLuc-ON promoter reporter clone | Genecopoeia | Cat# HPRM15772 | |
| Antibody | Human CXCL8/IL-8 neutralizing antibody | R and D | Cat# MAB208 | In vitro/in vivo chemotaxis experiments 2.8 ng/µl used for neutralization |
| Antibody | Rabbit anti-phospho-ERK3 (pSer189) | Sigma | Cat# SAB4504175 | WB 1:500 |
| Antibody | Rabbit anti-ERK3 | Cell Signaling Technology | Cat# 4067 | WB 1:500 |
| Antibody | Rabbit anti- MK5/MAPKAPK5 (D70A10) | Cell Signaling Technology | Cat# 7419 | WB 1:500 |
| Antibody | Rabbit anti-V5-tag | Cell Signaling Technology | Cat# 13202 | WB 1:500 |
| Antibody | Rabbit anti-p44/42 MAPK (ERK1/2) | Cell Signaling Technology | Cat# 9102 | WB 1:1000 |
| Antibody | Rabbit anti-phospho-p44/42 MAPK (ERK1/2) (Thr202/Tyr204) | Cell Signaling Technology | Cat# 9101L | WB 1:1000 |
| Antibody | Rabbit anti-phospho-p38 MAPK (Thr180/Tyr182) | Cell Signaling Technology | Cat# 9215 | WB 1:500 |
| Antibody | Rabbit anti-p38 MAPK antibody | Cell Signaling Technology | Cat# 9212 | WB 1:500 |
| Antibody | Rabbit anti-IκBα (44D4) | Cell Signaling Technology | Cat# 4812 | WB 1:500 |
| Antibody | Rabbit anti-phospho-SAPK/JNK (183/Y185) | Cell Signaling Technology | Cat# 9251 | WB 1:500 |
| Antibody | Normal Rabbit IgG | Cell Signaling Technology | Cat# 2729 | Used as a control for IP |

*Continued on next page*

*Continued*

| Reagent type (species) or resource | Designation | Source or reference | Identifiers | Additional information |
|---|---|---|---|---|
| Antibody | Rabbit anti-c-Jun (60A8) | Cell Signaling Technology | Cat# 9165 | WB 1:500<br>IF 1:400 |
| Antibody | Rabbit anti-Histone H3 (D1H2) | Cell Signaling Technology | Cat# 4499 | WB 1:500 |
| Antibody | Mouse anti-M2-PK antibody | Schebo Biotech AG | Cat# S-1 | WB 1:500 |
| Antibody | Rabbit anti-MAPK4 (ERK4) | Abcam | Cat# ab211501 | WB 1:500 |
| Antibody | Rabbit anti-PRAK/MK5 (phospho T182) antibody | Abcam | Cat# ab138668 | WB 1:500 |
| Antibody | Anti-β-actin HRP conjugated | Abcam | Cat# ab49900 | WB 1:40 000 |
| Antibody | Mouse, anti-Keratin 20 (Krt20) | Agilent Dako | Cat# M701929 | WB 1:1000 |
| Antibody | Mouse anti-GAPDH antibody | GeneTex | Cat# GTX627408 | WB 1:1000 |
| Antibody | Mouse anti-α-tubulin antibody | GeneTex | Cat# GTX628802 | WB 1:1000 |
| Antibody | Mono- and polyubiquitin conjugates monoclonal HRP-coupled antibody (FK2) | Enzo | Cat# BML-PW8810 | WB 1:250 |
| Antibody | HRP-conjugated secondary antibody for rabbit IgG | Invitrogen | Cat# A16096 | WB 1:40 000 |
| Antibody | HRP-conjugated secondary antibody for rabbit IgG | Invitrogen | Cat# 32460 | WB 1:2000 |
| Antibody | HRP-conjugated secondary antibody for mouse IgG | GE Healthcare Life Sciences | Cat# NA9310 | WB 1:20 000 |
| Antibody | Anti-ERK3 | R and D | Cat# MAB3196 | IF 1:400 |
| Antibody | Goat anti-rabbit IgG-Alexa 488 | Thermo Fisher Scientific | Cat# A11008 | IF, working concentration: 5 µg/ml |
| Antibody | Goat anti-mouse IgG-Cyanine3 | Thermo Fisher Scientific | Cat# A10521 | IF, working concentration:<br>2D cultures: 5 µg/ml<br>3D cultures: 8 µg/ml |
| Chemical compound, drug | DNA dye Hoechst 33342 | Thermo Fisher Scientific | Cat# H3570 | IF, working concentration: 10 µg/ml |
| Sequence-based reagent | qRT-PCR primers human ERK3 | Sigma | Frw_5' ATGGATGAGCCAATTTCAAG<br>Rv_5' CTGACAATCATGATACCTTTCC | |
| Sequence-based reagent | qRT-PCR primers human *CXCL8*#1 | Sigma | Frw_5' GAGCACTCCATAAGGCACAAA<br>Rv_5' ATGGTTCCTTCCGGTGGT | |
| Sequence-based reagent | qRT-PCR primers human *CXCL8*#3 | Sigma | Frw_5' TGTAAACATGACTTCCAAGC<br>Rv_5' AAAACTGCACCTTCACAC | |
| Sequence-based reagent | qRT-PCR primers human *IL16* | Sigma | Frw_5' CAGTGTTAATCCCTATTGCAC<br>Rv_5' ATTGTTGAGAGAGGGACTTC | |
| Sequence-based reagent | qRT-PCR primers human *CXCL6* | Sigma | Frw_5' CCTCTCTTGACCACTATGAG<br>Rv_5' GTTTTGGGGTTTACTCTCAG | |
| Sequence-based reagent | qRT-PCR primers human *TLR4* | Sigma | Frw_5' TGGAGGTGTGAAATCCAG<br>Rv_5' CTTGATAGTCCAGAAAAGGC | |

*Continued on next page*

*Continued*

| Reagent type (species) or resource | Designation | Source or reference | Identifiers | Additional information |
|---|---|---|---|---|
| Sequence-based reagent | qRT-PCR primers housekeeping human 18 s | Sigma | Frw_5' AGAAACGGCTACCACATCCA Rv_5' CACCAGACTTGCCCTCCA | |
| Sequence-based reagent | qRT-PCR primers housekeeping human GAPDH | Sigma | Frw_5' CGACAGTCAGCCGCATCTT Rv_5' CCCCATGGTGTCTGAGCG | |
| Sequence-based reagent | qRT-PCR primers human GKN1 | Sigma | Frw_ 5' agctcctgccctagctaactataa Rv_ 5' ttgtgttcattgttgacactcact | Used for HGOs qRT-PCR experiments |
| Sequence-based reagent | qRT-PCR primers human ERK3 | Sigma | Frw_ 5' tcgatgagtcggagaagtcc Rv_ 5' gaagatgtctttgttagtgatcaggt | Used for HGOs qRT-PCR experiments |
| Sequence-based reagent | qRT-PCR primers mouse Alpi | Sigma | Frw_ 5' AGGATCCATCTGTCCTTTGGT Rv_ 5' TTCAGCTGCCTTCTTGTTCC | Used for MCOs qRT-PCR experiments |
| Sequence-based reagent | qRT-PCR primers mouse Krt20 | Sigma | Frw_ 5' agtcccacctcagcatgaa Rv_ 5' gagctcagcatctcctggat | Used for MCOs qRT-PCR experiments |
| Sequence-based reagent | qRT-PCR primers mouse Erk3 | Sigma | Frw_ 5' acgacatgactgagccacac Rv_ 5' TCTGCTCCAGGAAATCCAGT | Used for MCOs qRT-PCR experiments |
| Sequence-based reagent | qRT-PCR primers housekeeping mouse Gapdh | Sigma | Frw_ 5' GTGCCAGCCTCGTCC Rv_ 5' ACCCCATTTGATGTTAGTGG | Used for MCOs qRT-PCR experiments |
| Software, algorithm | ImageJ | RRID:SCR_003070 | RRID:SCR_003070 https://imagej.net/ | Used for WB quantification and IF staining analyses |
| Software, algorithm | ImageJ Coloc2 Plugin | Self-modified version as described by French et al. (2008). | Self-modified version of ImageJ RRID:SCR_003070 | Used to quantify Fluorescence co-localizations c-Jun and ERK3 |
| Software, algorithm | Fiji | RRID:SCR_003070 | Fiji (RRID:SCR_002285) http://fiji.sc | Used for IF images visualization |

## Cell culture

HT-29 (ATCC HTB-38) cells and Human Colonic Primary Epithelial Cells (HCPECs) (CCD 841 CoN, ATCC–CRL-1790) were purchased from ATCC (Manassas, VA 20108 USA) and were passaged until passage six. HT-29 cells were cultured in McCoy's medium supplemented with 10% heat inactivated Fetal Bovine Serum (FBS). HCPECs were cultured in Minimum Essential Medium Eagle's (MEM) supplemented with 10% heat inactivated FBS, 2 mM L-glutamine and 1 mM sodium pyruvate. The authenticated CaCo2 cell line was a kind gift from Prof. Monilola Olayioye (University of Stuttgart) and was cultured in Roswell Park Memorial Institute (RPMI) medium supplemented with 10% heat inactivated FBS. MDA-MB231 (ACC 732) cells and THP1 monocytic cell line (ACC 16) were purchased from DSMZ and cultured in Dulbecco's Modified Eagle Medium (DMEM)/RPMI medium supplemented with 10% heat inactivated FBS, respectively. 293 T cells were a kind gift from Dr. Andreas Ernst (Goethe-University Frankfurt am Main, IBC2) and were cultured in DMEM supplemented with 10% FBS.

Cells used in this study were authenticated cell lines obtained from ATCC or DSMZ. All used cells were periodically tested for *Mycoplasma* contamination with negative results.

## Stimulation of cells

Cells were seeded in 12-well plates at an initial density of $2 \times 10^5$ cells/well. Next day medium was exchanged for either serum free medium (HT-29, CaCo2) or MEM minus FBS, L-glutamine and sodium pyruvate (HCPECs). Cells were stimulated with 200 ng/ml LPS (Cat# L6143, Sigma, 1 mg/ml) at indicated time points. Human recombinant IL-1β (Cat# 11340013, ImmunoTools), Pam3CSK4 TLR1/2 ligand (Cat# tlrl-pms, InvivoGen) and R848 TLR7/8 ligand (Cat# tlrl-r848, InvivoGen) were used at 10 ng/ml, 20 μg/ml and 2.5 μg/ml, respectively at indicated time points.

## Cycloheximide chase experiments

To investigate the half-life of ERK3, Cycloheximide (CHX) chase experiments were performed. Cells were seeded in 12-well plates at an initial density of $2 \times 10^5$ cells/well. Next day, the medium was exchanged to FBS free medium and protein biosynthesis was inhibited by treatment with 100 µg/ml of CHX (Cat# C-7698, Sigma, stock 100 mg/ml in DMSO) for indicated time points. If LPS pre-treatment was included in the experiment, cells were stimulated for 30 min with LPS (200 ng/ml) prior to the treatment with CHX. Western blot analyses were performed and fold change in ERK3 protein levels was calculated in respect to the untreated cells (-LPS, 0 hr) or to the respective control in each group (0 hr) for unstimulated (-LPS) and LPS stimulated (+LPS) cells using ImageJ software.

## Chemical inhibitors

To investigate a possible role for the proteasome in regulation of ERK3 protein, MG-132 inhibitor (Calbiochem, Cat# 474790, Merck Millipore) was used at a final concentration of 10 µM for 6 hr. Selective MEK1/2 inhibitor trametinib (GSK1120212, Cat# S2673, Selleckchem) was used at a working concentration of 1 µM for 1 hr, before LPS stimulation. DMSO (Cat# A3672.0250, Applichem) was used as a solvent control for both inhibitors.

## Generation of knockdowns

All shRNAs plasmids were purified from the MISSION shRNA Human Library (Sigma) following manufacturer's instructions. As control the non-targeting control shRNA (shCo) MISSION pLKO.1 puro (Cat# SHC001) was included.

> shRNA targeting ERK3
> shMAPK6#1 (shERK3#1) TRCN0000001568, NM_002748.x-3734s1c1,
> CCGGGCTGTCCACGTACTTAATTTACTCGAGTAAATTAAGTACGTGGACAGCTTTTT
> shMAPK6#2 (shERK3#2) TRCN0000001569, NM_002748.x-1564s1c1, CCGGGACATGACTGAGCCACACAAACTCGAGTTTGTGTGGCTCAGTCATGTCTTTTT
> shMAPK6#3 (shERK3#3) TRCN0000001570, NM_002748.x-798s1c1,
> CCGGTGATCTGGGTTCTAGGTATATCTCGAGATATACCTAGAACCCAGATCATTTTT
> shRNA targeting MK5 shMAPKAPK5#1 (shMK5#1) TRCN0000194823, NM_003668.2–475 s1c1
> CCGGCCCAAACATAGTTCAGATTATCTCGAGATAATCTGAACTATGTTTGGGTTTTTTG
> shMAPKAPK5#5 (shMK5#5) TRCN0000000682, NM_003668.x-1622s1c1
> CCGGGAAATTGTGAAGCAGGTGATACTCGAGTATCACCTGCTTCACAATTTCTTTTT
> shRNA targeting ERK4
> shMAPK4#1 (shERK4#1) TRCN0000001374, NM_002747.x-3808s1c1
> CCGGCTCACACCACACGCCTTAAATCTCGAGATTTAAGGCGTGTGGTGTGAGTTTTT
> shMAPK4#2 (shERK4#1) TRCN0000001375, NM_002747.x-1105s1c1
> CCGGACTACACCAAAGCCATCGACACTCGAGTGTCGATGGCTTTGGTGTAGTTTTT
> shMAPK4#5 (shERK4#5) TRCN0000001378, NM_002747.x-1017s1c1
> CCGGGATCAGCATTACTCCCACAAGCTCGAGCTTGTGGGAGTAATGCTGATCTTTTT

> siRNAs directed against ERK3 were purchased from Qiagen.
> siMAPK6#1 (siERK3#1) FlexiTube siRNA 5 nmol, siRNA Name: Hs_MAPK6_5, Cat# SI00606025
> Sense strand: 5'- AGUUCAAUUUGAAAGGAAATT-3'
> siMAPK6#2 (siERK3#2) FlexiTube siRNA 5 nmol, siRNA Name: Hs_MAPK6_6, Cat# SI00606032

(siRNA used only in *Figure 4—figure supplement 2B*)

> Negative control siRNA (siCo) Cat# 1027310.

Cells were seeded one day before transfection at an initial density of $2 \times 10^5$ cells/well in 12-well plates or at $3 \times 10^5$ cells/well (6-well plates). Cells were transfected using SAINT-sRNA transfection reagent (SR-2003, Synvolux) according to the manufacturer's instructions. Unless otherwise indicated, medium was exchanged 24 hr post-transfection for medium without FBS and 48 hr post–transfection supernatants were harvested for IL-8 levels measurement by ELISA. In addition, the knockdown was verified by Western blot and/or quantitative real-time PCR.

## CRISPR/Cas9 constructs

CRISPR/Cas gRNA sequences targeting ERK3 were designed by Rule Set 2 of Azimuth 2.0 as described previously (*Doench et al., 2016*). The top three scoring gRNAs were selected:

#1 5'-CACCGAGCCAATTAACAGACGATGT-3'
#2 5'-CACCGATACTTGTAACTACAAAACG-3'
#3 5'-CACCGCTGCTGTTAACCGATCCATG-3' gRNAs were individually cloned into pLenti-CRISPRv2 (Addgene plasmid #52961), following established protocols (*Sanjana et al., 2014*).

## Lentiviral-mediated expression of cDNAs

Human full-length wild type ERK3 (ERK3 WT) cDNA was purified from Human Kinase Library (Addgene). ERK3 WT pDONR-223 was used as a template to generate ERK3 K49A K50A kinase dead mutant. Site-directed mutagenesis was implemented by PCR using Q5 High-Fidelity DNA Polymerase (Cat# MO491, New England BioLabs) and the following primers were used: frw_5' GCAATTG TCCTTACTGATCCCCAGAGTGTC, rev_5' CGCGATGGCTACTCTTTGTCACAGTC. For lentiviral expression, ERK3 WT and ERK3 K49A K50A mutant in pDONR-223 were transferred into the destination vector pLenti4TO/V5-Dest by clonase reaction (Gateway LR Clonase II Enzyme Mix Cat# 11791–020, ThermoFisher Scientific). ERK3 WT and ERK3 K49A K50A mutant were also cloned into pcDNA3/V5-Dest40 vector for transient transfection purposes. For the production of lentiviral particles, the following packaging plasmids were used: pHDM-G (encoding VSV-G), pHDM Hgpm2 (encoding codon-optimized HIV gag-pol proteins), pHDM tat 1b (encoding HIV Tat1b protein) and pRC CMV-Rev1b (encoding HIV rev protein). Lentiviral particles were generated following standard protocols. In short, lentiviral supernatants were produced in 293 T cells by co-transfection of the cells with lentiviral packaging plasmids (0.3 µg each), lentiviral expression constructs (1 µg) and 10.8 µl of 10 mM polyethylenimine (PEI). The viral particles were harvested after 48 hr and sterile-filtered. HT-29, MDA-MB231 and HeLa cells were infected with lentiviral particles in the presence of 10 µg/ml of polybrene (Cat# sc-134220, Santa Cruz). Cells were then selected with puromycin (Cat# 0240.3, Carl Roth) at following concentrations: 3 µg/ml (MDA-MB231, HeLa) or 8 µg/ml for HT-29, until a stable knockdown was achieved. For complementation assays, empty vector control (pLenti4TO/V5-Dest), wild type or kinase dead (K49A K50A) mutant of ERK3 were reintroduced into shERK3 (3'UTR) background by lentiviral transduction and cells were double-selected with zeocin (100 µg/ml) (Cat# R25001, Invitrogen) and puromycin.

To generate CRISPR/Cas9 mediated ERK3 knockout in the HT-29 cell line, cells were infected with lentiviral particles and selected with puromycin (30 µg/ml). Lentiviral particles coding for CRISPR ERK3 (CRISPR ERK3) and CRISPR control vector (pLentiCRISPRv2) (CRISPR Co) were produced in 293 T cells by co-transfection of lentiviral packaging plasmids (0.3 µg each) and 1.1 µg of lentiviral vector containing the respective gRNAs in the presence of 21 µl of Lipofectamine2000 (Cat# 11668027, ThermoFisher Scientific).

## Transient transfections

HeLa cells stably transfected with shRNA targeting ERK3 at 3'UTR (shERK3) or with control empty vector shRNA (shCo) were transiently transfected with either an empty vector pcDNA3/V5-Dest40 (EV), ERK3 WT or ERK3 K49A K50A mutant construct (0.5 µg plasmid) in the presence of Lipofectamine2000 (Cat# 11668027, ThermoFisher Scientific) (3 µl/well). 6 hr post-transfection medium was exchanged for DMEM + FBS complete medium. 24 hr post-transfection medium was exchanged again for DMEM–FBS medium. 48 hr post-transfection supernatants were harvested for IL-8 ELISA and cells were analyzed by western blot.

## Enzyme-linked immunosorbent assay (ELISA)

Secreted protein concentration of IL-8 was measured by ELISA. The assay was performed according to manufacturer's instructions (Human IL-8 ELISA Ready-SET-Go! Kit, Cat# 88–8086, eBioscience).

## Cell fractionation

Cell fractionation experiments were performed using the Minute Plasma Membrane Protein Isolation and Cell Fractionation Kit (Cat# SM-005, Invent Biotechnologies) according to the manufacturer's instructions. Histone H3 was used as a control for the nuclear fraction and M2-PK as a control for the cytosolic fraction.

## Isolation of human neutrophils

Neutrophils were prepared from heparinized peripheral blood obtained from healthy volunteers. For dextran sedimentation blood was mixed with 3% dextran 500 (Cat# 9219.1, Carl Roth) in 0.9% NaCl, at a 1:1 ratio (7.5 ml blood in 7.5 ml dextran solution) by gentle inverting prior to 20–30 min incubation at room temperature (RT). The layers containing neutrophils were harvested; 6 ml of the leukocyte-rich layer was gently pipetted onto 7.5 ml of Histopaque-1077 (Cat# 10771, Sigma) and low-density gradient centrifugation was performed at 1700 rpm, at RT for 30 min. Supernatants were then removed and pellets were gently resuspended in 7.5 ml of ACK lysing buffer, samples were incubated for 10 min at RT, protected from light, followed by 2 min centrifugation at 1700 rpm. To maintain clean populations of leukocytes with no residual erythrocytes, samples were washed with phosphate buffered saline (PBS), pH 7.2. Neutrophil cell pellets were resuspended in MEM, viability and cell number was assessed.

## Neutrophils and THP1 in vitro chemotaxis - transwell migration assay

The effects of ERK3-dependent IL-8 levels present in cell culture supernatants from HCPECs and HT-29 cells on neutrophils and THP1 chemotaxis was assessed by transwell migration assays. 6.5 mm transwell inserts with either 5.0 μm (neutrophils) or 8 μm (THP1 cells) pore polycarbonate membrane (Cat# 3421/3422, respectively, Corning) were used. In the lower chamber of the transwell plate, HCPECs were seeded and transiently infected with either a control shRNA (shCo) or an shRNA targeting ERK3. Alternatively, supernatants obtained from HCPECs transfected with either a negative control shRNA (shCo)/siRNA (siCo) or shRNA/siRNA targeting ERK3 (ERK3 KD) were placed in the lower chamber. Freshly isolated neutrophils obtained from peripheral blood of healthy volunteers were pre-stained with 5 μM CellTracker Green CMFDA (Cat# C7025, ThermoFisher Scientific) for 15 min prior migration, followed by 30 min stimulation with LPS (200 ng/ml). Neutrophils were added to the inserts (5 μm pore size) at a final concentration of $3 \times 10^5$ cells per insert. To asses THP1 cells chemotaxis, HT-29 cells carrying anti ERK3 stable knockdown (shERK3) or empty vector control cells (shCo) were seeded in the lower chamber alternately with supernatants obtained from HCPECs transfected with either a control vector (shCo)/negative control siRNA (siCo) or shRNA/siRNA targeting ERK3 (ERK3 KD). THP1 cells were counted and stained with 5 μM of CellTracker Green CMFDA for 15 min, cells were then resuspended in 200 μl of RPMI medium with no FBS and $1.2 \times 10^5$ cells were added into each insert (8 μm pore size). To determine the role of IL-8 in the observed chemotaxis, human CXCL8/IL-8 neutralizing antibody (Cat# MAB208, R and D) was used in the lower compartment at a concentration of 2.8 ng/μl. Following 2 hr incubation at 37°C, migration of neutrophils or THP1 to the lower chambers was measured using fluorescence (excitation wavelength 480 nm, emission wavelength 535 nm). Fold change of Relative Fluorescence Units (RFU) was then calculated for each condition. Medium control was incorporated into each experiment to determine background rate RFU, which were then subtracted from all tested conditions.

## Animal experiments

### In vivo studies of murine leukocytes chemotaxis

To assess the physiological impact of ERK3-depletion from human intestine epithelial cells, an in vivo chemotaxis experiment was performed. All animal experiments were approved by local authorities (National Investigation Office Rheinland-Pfalz, Approval ID: AZ 23 177–07/G17-1-036) and conducted according to the German Animal Protection Law. Eight-week-old female C57BL/6J mice were purchased from Janvier Labs. HCPECs were seeded in six-well plate at an initial density of $3 \times 10^5$ cells/well. After 24 hr, cells were transfected with either a negative control siRNA (siCo) or siRNA specific to ERK3 (siERK3) in the presence of Saint-sRNA transfection reagent. 24 hr post-transfection medium was exchanged for 10 ml of MEM without FBS, L-glutamine and sodium pyruvate. After 24 hr, supernatants were harvested and further concentrated using Amicon Ultra-15 Centrifugal Filter (3 kDa, Cat# UFC900324, Merck Millipore). IL-8 concentration was determined by ELISA.

Groups of 5 eight-week-old C57BL/6J female mice were injected intraperitoneally (i.p) with one of the following: 1 ml of MEM without supplements (MEM control), 1 ml of MEM containing 900 ng of human recombinant CXCL8/IL-8 (Cat# 200–08, Peprotech) (rhCXCL8/IL-8), 1 ml of HCPECs siCo concentrated supernatant or 1 ml of HCPECs siERK3 concentrated supernatant. 4 hr post-injections mice were sacrificed and peritoneal white blood cells populations were harvested by peritoneal

lavage with 10 ml of cold PBS pH 7.2, supplemented with 1% heat-inactivated FBS. Cell suspensions were centrifuged at 1300 rpm for 5 min. Pellets were resuspended in 100 µl of PBS and transferred into 1.5 ml Eppendorf tube. Number of white blood cells (WBC) per $mm^3$ and the percentage of granulocytes, monocytes and lymphocytes was measured with scil Vet abc hematology counter. Absolute cell number per $mm^3$ was then calculated using formula: WBC (total number) x (% of leukocyte/100).

## Lung metastasis model

For the lung seeding and metastasis model, 10-week-old female NOD.CB17-$Prkdc^{scid}$/J mice were injected intravenously (i.v) with $5 \times 10^5$/100 µl of control (shCo) or ERK3 knockdown (shERK3) MDA-MB231 cells. After 7 weeks, animals were sacrificed and lungs were dissected. After fixation in 4% paraformaldehyde (PFA) lungs were embedded in paraffin for sectioning. The animal experiment was performed under the permission (G16-1-026) of the National Investigation Office Rheinland-Pfalz and conducted according to the German Animal Protection Law.

## Histology and lesions quantification

After dissection and fixation of lung lobes in 4% PFA at 4°C, dehydration and paraffinization was performed and 4-µm-thick sections were produced every 100 µm. Sections were stained with Hematoxylin and Eosin (H and E) in the Immunohistochemistry FZI core facility, University Medical Center of the JGU Mainz). H and E-stained sections (five sections per lung) were photographed with a Nikon D90 digital camera. Using color deconvolution, micrometastases in the lungs were analyzed using ImageJ software. The number of tumors in each section was assessed and metastatic burden was calculated by dividing total tumor area by total tissue area of the analyzed section and was expressed as percentage.

## Antibodies

Anti-phospho-ERK3 (pSer189) antibody (Cat# SAB4504175) was purchased from Sigma. Anti-ERK3 antibody (Cat# 4067), anti-MK5/MAPKAPK5 (D70A10) antibody (Cat# 7419), anti-V5-tag antibody (Cat# 13202), anti-p44/42 MAPK (ERK1/2) antibody (Cat# 9102), anti-phospho-p44/42 MAPK (Thr202/Tyr204) antibody (Cat# 9101L), anti-phospho-p38 MAPK (Thr180/Tyr182) antibody (Cat# 9215), anti-p38 MAPK antibody (Cat# 9212), anti-IκBα (44D4) antibody (Cat# 4812), anti-phospho-SAPK/JNK (183/Y185) antibody (Cat# 9251), Normal Rabbit IgG antibody (Cat# 2729), anti-c-Jun (60A8) antibody (Cat# 9165) and Histone H3 (D1H2) antibody (Cat# 4499) were purchased from Cell Signaling Technology (Danvers, MA). Anti-M2-PK antibody (S-1) was purchased from Schebo Biotech AG. Anti-MAPK4 (ERK4) antibody (Cat# ab211501), anti-PRAK/MK5 (phospho T182) antibody (Cat# ab138668) and anti-β-actin HRP conjugated antibody (Cat# ab49900) were purchased from Abcam. Anti-Keratin 20 (Krt20) antibody (Cat# M701929) was purchased from Agilent Dako. Anti-GAPDH antibody (Cat# GTX627408) and anti-α-tubulin antibody (Cat# GTX628802) were purchased from GeneTex. Mono- and polyubiquitin conjugates monoclonal HRP-coupled antibody (FK2) (Cat# BML-PW8810) was purchased from Enzo. HRP-conjugated secondary antibodies for rabbit IgG were obtained from Invitrogen (Cat# A16096) and (Cat# 32460) and secondary antibody for mouse IgG from GE Healthcare Life Sciences (Cat# NA9310).

Anti-ERK3 antibody (Cat# MAB3196) used for immunofluorescence staining was purchased from R and D and was independently validated by immunohistochemistry of control and ERK3 knockdown cells as well as by western blot analysis of control and ERK3-depleted HT-29 cells used for immunofluorescence staining (*Figure 7B*). Secondary goat anti-rabbit IgG-Alexa 488 (Cat# A11008) and secondary goat anti-mouse IgG-Cyanine3 (Cat# A10521) were purchased from Thermo Fisher Scientific.

## Western blotting

Cells were washed with ice-cold PBS (10 mM sodium phosphate, 150 mM NaCl, pH 7.2) and lysed in cold RIPA lysis buffer (250 mM NaCl, 50 mM Tris (pH 7.5), 10% glycerin, 1% Triton X-100), supplemented with protease inhibitor cocktail Set I-Calbiochem 1:100 (Cat# 539131, Merck Millipore) and phosphatase inhibitors (1 mM sodium orthovanadate ($Na_3VO_4$), 1 mM sodium fluoride (NaF). Cells were lysed for 30 min on ice, followed by 10 min centrifugation at 14,000 rpm. Protein concentrations were estimated determined using 660 nm Protein Assay (Cat# 22660, Thermo Fisher Scientific).

Samples were prepared by mixing with 4xSDS-PAGE sample buffer (277.8 mM Tris-HCl pH 6.8; 44.4% glycerol, 4.4% SDS, 0.02% bromophenol blue) supplemented with 50 mM DTT per ml of sample buffer. After boiling at 95℃ for 5 min samples were subjected to 7.5%/10% SDS-PAGE followed by transfer of the proteins onto nitrocellulose membranes (GE Healthcare, Chalfont St Giles, UK). Membranes were blocked in 3% BSA/PBST (1x PBS, pH 7.2 containing 0.05% Tween-20) for 1 hr at RT. Membranes were then washed 3 × 5 min with PBST and incubated with primary antibody diluted in PBST at 4℃, overnight. Following 3 × 5 min washing with PBST membranes were incubated with HRP-conjugated secondary antibody for 1 hr at RT. After washing, signal was visualized using chemiluminescent HRP substrate (Immobilon Western, Cat# WBKLS0500, Merck Millipore). Western blot semi-quantification was performed using ImageJ software.

## Immunoprecipitation assays (IP)

### Endogenous ubiquitination of ERK3 protein

For immunoprecipitation of endogenous ERK3, HT-29 and HCPECs cells were seeded in 10 cm dishes at an initial density of $2 \times 10^6$ cells per dish. Next day, medium was exchanged for medium without FBS and supplements and cells were treated with MG-132 inhibitor for 6 hr prior to LPS (200 ng/ml) stimulation for 4 hr. Afterwards, medium was aspirated, cells were washed with ice-cold PBS and lysed with ice-cold IP buffer (10 mM HEPES pH 7.4; 150 mM NaCl, 1% Triton X-100, plus protease inhibitor cocktail Set I-Calbiochem 1:100 (Cat# 539131, Merck Millipore), 1 mM $Na_3VO_4$ and 1 mM NaF). After 30 min on ice, samples were centrifuged at 14000 rpm for 10 min, followed by protein concentration measurement using 660 nm Protein Assay (Cat# 22660, ThermoFisher). Antibody-protein complexes were precipitated by Protein A/G-Agarose beads (Cat# 11 134 515 001/11 243 233 001, Roche). Beads were washed twice with 200 µl of IP buffer and lysates were added along with the ERK3 antibody. The mix was incubated for 2 hr at 4℃ with rotation. After the incubation, beads were washed three times with 500 µl of IP buffer, centrifuged each time for 30 s at 1000 rpm, sample buffer was added and samples were boiled for 5 min at 95℃. SDS-PAGE was performed.

### Pull down of endogenous c-Jun/ERK3

HT-29 cells were stimulated with LPS (200 ng/ml) for 4 hr in medium without FBS. After the stimulation, cells were washed with ice-cold PBS and lysed with ice-cold IP buffer (10 mM HEPES pH 7.4; 150 mM NaCl, 1% Triton X-100, plus protease inhibitor cocktail Set I-Calbiochem 1:100 (Cat# 539131, Merck Millipore), 1 mM $Na_3VO_4$ and 1 mM NaF). After 30 min on ice, samples were centrifuged at 14,000 rpm for 10 min. Protein A/G-Agarose beads (Cat# 11 134 515 001/11 243 233 001, Roche), lysates and either c-Jun, ERK3 or Normal Rabbit IgG antibody were incubated for 2 hr at 4℃ with rotating. After the incubation, beads were washed with IP buffer and analyzed by immunoblot.

### RNA isolation, cDNA synthesis and RT-PCR analysis

For gene expression analyses, cells were washed with cold PBS and total RNA was extracted using Trizol (Cat# 15596018, Ambion) according to the manufacturer's instructions. Quality of the RNA was evaluated by NanoDrop (ThermoFisher Scientific), absorbance at 260/280 was measured and samples within range of 2.0 ± 0.3 were used. Isolated RNA (500 ng) was then used as a template for cDNA synthesis with the RevertAid First Strand cDNA synthesis kit (Cat# K1621, ThermoFisher Scientific) and random hexamer primers.

Real-time PCR was performed using EvaGreen qPCR master mix (5 x Hot Start Taq EvaGreen qPCR Mix (No ROX), Cat# 27490, Axon) and following primers:

*ERK3* Frw_5' ATGGATGAGCCAATTTCAAG
Rv_5' CTGACAATCATGATACCTTTCC
*CXCL8*#1 Frw_5' GAGCACTCCATAAGGCACAAA
Rv_5' ATGGTTCCTTCCGGTGGT,
*CXCL8*#3 Frw_5' TGTAAACATGACTTCCAAGC
Rv_5' AAAACTGCACCTTCACAC;
*IL16* Frw_5' CAGTGTTAATCCCTATTGCAC
Rv_5' ATTGTTGAGAGAGGGACTTC;
*CXCL6* Frw_5' CCTCTCTTGACCACTATGAG
Rv_5' GTTTTGGGGTTTACTCTCAG.

*TLR4* Frw_5' TGGAGGTGTGAAATCCAG
Rv_5' CTTGATAGTCCAGAAAAGGC.
The housekeeping genes for human 18S or GAPDH were used for normalization:
*18* s Frw_5' AGAAACGGCTACCACATCCA
Rv_5' CACCAGACTTGCCCTCCA;
*GAPDH* Frw_5' CGACAGTCAGCCGCATCTT
Rv_5' CCCCATGGTGTCTGAGCG
Human gastric organoids primers
*GKN1* Frw_ 5' agctcctgccctagctaactataa
Rv_ 5' ttgtgttcattgttgacactcact
*ERK3* Frw_ 5' tcgatgagtcggagaagtcc
Rv_ 5' gaagatgtcttttgttagtgatcaggt
Mouse colon organoids primers:
*Alpi* Frw_ 5' AGGATCCATCTGTCCTTTGGT
Rv_ 5' TTCAGCTGCCTTCTTGTTCC
*Krt20* Frw_ 5' agtcccacctcagcatgaa
Rv_ 5' gagctcagcatctcctggat
*Erk3* Frw_ 5' acgacatgactgagccacac
Rv_ 5' TCTGCTCCAGGAAATCCAGT
Mouse *Gapdh*
*Gapdh* Frw_ 5' GTGCCAGCCTCGTCC
Rv_ 5' ACCCCATTTGATGTTAGTGG

Relative expression levels were calculated as $\Delta\Delta Ct$ and results are presented as log2fold change in gene expression.

## Secretome analysis by L-series human antibody array 1000

HCPECs were seeded in 12-well plates at an initial density of $2 \times 10^5$ cells/well. 24 hr later cells were transiently transfected with either control siRNA (siCo) or siRNA targeting ERK3 (siERK3). 48 hr post-transfection medium was exchanged for MEM-FBS and other supplements. 24 hr later supernatants were harvested from each well for secretome analysis. Cells were lysed in RIPA buffer and total protein concentrations were measured using 660 nm Protein Assay (Cat# 22660, Thermo Fisher Scientific), cells were subjected for immunoblot analysis to determine knockdown efficiency.

## Antibody array

The RayBiotech human L-Series biotin-based antibody array was purchased from Tebu-Bio (Cat# AAH-BLG-1000) and performed according to the manufacturer's instructions. Briefly, supernatants were dialyzed prior biotin labeling. Labeled proteins were then incubated on the blocked glass slides at RT. Two slides were provided: L-507 and L-493 coated with indicated numbers of capture antibodies. Array slides were subsequently washed and fluorescence (Cy3) label-conjugated Streptavidin was added. Slides were then dried and sent for fluorescence detection and analysis by RayBio Software.

## RNA sequencing and bioinformatic analyses

For transcriptome analysis three biological replicates of HCPECs were seeded in 12-well plates at an initial density of $2 \times 10^5$ cells/well. 24 hr later cells were transiently transfected with either control siRNA (siCo) or siRNA targeting ERK3 (siERK3). 24 hr post-transfection medium was exchanged for MEM-FBS and other supplements and cells were stimulated with LPS (200 ng/ml) for 24 hr. After the stimulation, cells were washed with cold PBS and lysed with Trizol (Cat# 15596018, Ambion) according to the manufacturer's instructions. Total RNA was quantified by a Qubit 2.0 fluorometer (Invitrogen). Quality was assessed using Agilent's bioanalyzer 2100 and a RNA 6000 Nano chip (Agilent). Samples with RNA integrity number (RIN) >8 were further subjected for RNA library preparation. Barcoded cDNA libraries were prepared from 300 ng of total RNA using the NEBnext Poly(A) mRNA Magnetic Isolation Module and NEBNext Ultra RNA Library Prep Kit for Illumina (NEB) according to the manufacturer's instruction. Library quantity was assessed on a Qubit 2.0 using Qubit HS assay kit (Invitrogen). Library size was determined using Agilent's Bioanalyzer 2100 and a HS DNA assay chip. Barcoded RNA-Seq libraries were on board clustered using HiSeq Rapid SR Cluster Kit v2 using 8pM and 59 bps were sequenced on an Illumina HiSeq2500 using a HiSeq Rapid SBS kit v2.

Quality control on the sequencing data (59 base pairs, single end) was performed with the FastQC tool (available at http://www.bioinformatics.babraham.ac.uk/projects/fastqc/), as well as the comprehensive Qorts suite. By inspecting the produced reports, all samples were deemed of good quality and were further processed. Short reads alignment was performed with the ENSEMBL Homo_sapiens. GRCh38 was chosen as the reference genome. The corresponding annotation (ENSEMBL v79) was retrieved from the ENSEMBL FTP website (http://www.ensembl.org/info/data/ftp/index.html). STAR aligner (version 2.4.0b) was used to perform mapping to the reference genome (Dobin et al., 2013). Subsequent analyses were performed with R statistical software (version 3.5.0), leveraging core packages of the Bioconductor project. Alignments were processed with the 'featureCounts' function of the Rsubread package, using the annotation file also used for supporting the alignment. Exploratory data analysis and functional annotation to Gene Ontology terms was performed with the pcaExplorer package (version 2.6.0, Marini and Binder 2018: pcaExplorer: an R/Bioconductor package for interacting with RNA-seq principal components. BioRxiv. https://doi.org/10.1101/493551). Differential expression analysis was performed with the DESeq2 package (version 1.20.0), limiting the false discovery rate to 0.05 (Love et al., 2014). The apeglm (package version 1.2.1) shrinkage estimator was used to calculate the effect size for the contrasts of interest (Zhu et al., 2019). MA-plots were generated with the ideal package (version 1.4.0). Log2FC profiles for the different contrasts were plotted as heatmaps with the pheatmap package (version 1.0.12). Intersection between different sets are displayed in Venn diagrams, generated with the gplots package (version 3.0.1). Expression plots for selected genes display the individual values for the normalized counts, with a bar to show the median in each group.

## TF activation profiling plate array I (FA-1001)

HCPECs were seeded in 10 cm dishes at an initial density of $2 \times 10^6$ cells. After 24 h cells were transfected with either negative control siRNA (siCo) or siRNA targeting ERK3, using Saint-sRNA transfection reagent according to the manufacturer's instructions. 24 hr post-transfection medium was exchanged for MEM minus FBS and other supplements and cells were stimulated with LPS (200 ng/ml) for 24 hr. After stimulation, part of the cells was lysed in RIPA buffer for further western blot analysis and knockdown verification. Residual cells were subjected to nuclear extraction according to instructions provided with Nuclear Extraction Kit (Cat# SK-0001, Signosis).

TFs profiling array was performed following the instructions in the user manual provided by the manufacturer (Signosis). Briefly, 5 µg of nuclear extract was incubated with biotin-labeled probes. TF/probe complexes were then purified from the unbound probes and hybridized with the plate pre-coated with complementary sequences for each probe. Captured DNA probes were then incubated with Streptavidin-HRP conjugate, followed by substrate solution. Luminescence was then measured (integration time 1 s) and Relative Luminescence Units (RLU) are presented.

## Transcription factor filter plate assay

HCPECs were seeded in six-well plates at an initial density of $3 \times 10^5$ cells per well. After 24 hr, cells were transfected with either negative control siRNA (siCo) or siRNA targeting ERK3, using Saint-sRNA transfection reagent. 24 hr post-transfection cells were stimulated with LPS (200 ng/ml) for 24 hr in medium without any supplements.

Control (CRISPR Co/shCo) and ERK3 knockout (CRISPR ERK3) HT-29 cells were seeded in 6-well plates. Once the cells reach 70% to 80% confluence, medium was exchanged to FBS-free medium and cells were stimulated with LPS (200 ng/ml) for 24 hr. After stimulation, nuclear extracts were prepared by using the Nuclear Extraction Kit (Cat# SK-0001, Signosis) and samples were subjected to the Filter Plate Assay using AP-1 (Cat# FA-0004), C/EBP (PP-0011) or CREB (PP-0015) probe (Signosis).

The assay was performed according to the manufacturer's instructions. Briefly, nuclear extracts were incubated with biotin-labelled AP-1/CREB/C/EBP DNA-binding sequences in order to allow TF-DNA complex formation. AP-1/CREB/C/EBP-bound probes were then retained by the filter plate. Pre-labeled AP-1/CREB/C/EBP probes were eluted from the filter, followed by hybridization to 96-well hybridization plate. Captured AP-1/CREB/C/EBP probes were further detected with streptavidin-HRP and luminescence was measured as Relative Light Units (RLU). Fold change in RLU was calculated with respect to the control cells.

## CXCL8 promoter luciferase reporter assay

The CXCL8-Gaussia Luciferase GLuc-ON promoter reporter clone was purchased from Genecopoeia (HPRM15772) as a lentiviral expression construct along with the negative control plasmids (PEZX-LvPG02) with non-promoter sequence. MDA-MB231 cells were infected with lentiviral particle containing supernatant produced in 293 T cells as described before and selected with puromycin. Afterwards, stable cells were transiently transfected with siCo or siRNA targeting ERK3. 24 hr post-transfection, medium was exchanged for DMEM-FBS and cells were cultured for additional 24 hr. Supernatants were harvested and IL-8 promoter activity was assessed by measurement of secreted Gaussia Luciferase activity using Secrete-Pair Gaussia Luciferase Assay Kit (Cat# SPGA-G, Genecopoeia) and GL-H buffer with GL-substrate according to the manufacturer's instructions. Relative Luminescence Units (RLU) were measured (integration 1 s). For quantification, RLU values of negative control were subtracted from IL-8 promoter expressing samples and presented as fold change of siERK3 RLU normalized to siControl samples.

## Immunofluorescence

Control (shCo) and ERK3 knockdown (shERK3) HT-29 cells were seeded on coverslips. Next day, medium was exchanged to FBS-free medium and cells were stimulated with LPS (200 ng/ml) for 1.5 hr. After the treatment, cells were fixed in 3.7% formaldehyde (Cat# CP10.1, Roth) for 15 min, followed by washing with PBS and 3 min permeabilization using 0.1% Triton X-100 (AppliChem). After washing twice with PBS, cells were blocked with 1% BSA (Sigma) in PBS for 15 min and washed once with PBS. Staining was performed with anti-ERK3 antibody (Cat# MAB3196, R and D) and anti-c-Jun (60A8) (Cat# 9165, Cell Signaling) (dilution 1:400) antibody in blocking solution for 1 hr at RT. Afterwards, cells were washed with PBS and incubated with secondary antibodies: anti-rabbit IgG-Alexa 488 (Cat# A11008, ThermoFisher), secondary anti-mouse IgG-Cyanine3 (Cat# A10521, Thermo Fisher Scientific) at 5 µg/ml and DNA dye (Hoechst 33342) (Cat# H3570) at 10 µg/ml in blocking solution for 1 hr at RT in the dark. Samples were washed twice with PBS and cells were mounted onto glass slides using Moviol (+DABCO) (Sigma). Cells were imaged using a Leica DMi8 confocal microscope (63x, oil immersion objective).

## Image analyses

Confocal microscopy image processing was performed using ImageJ software. The mean fluorescence intensity for the nucleus region of interest (ROI) was calculated by subtracting background fluorescence from each image.

Fluorescence co-localizations within the nucleus were determined using a self-modified version of the ImageJ Coloc2 Plugin as described by *French et al. (2008)*. The calculated Pearson's rank indicates tendencies toward co-localization with a 100%, perfect co-localization with a score of 1 and complete separation at a score of −1. Scores above 0 indicate a tendency toward co-localization.

## Gastrointestinal organoids

### Mice

6–10 weeks old wild-type C57BL/6J mice were used for the experiment.

### Human tissue material

Normal human stomach organoids were derived from tumor adjacent normal stomach after tumor resection. Tissue samples used in this study were provided by the University Medical Center of the JGU Mainz. Written informed consent was obtained from all patients, and the study was approved by the ethical committee at the University Medical Center of the JGU Mainz (approval # 837.100.16 (10419).

### Mouse colon crypts and human gastric glands isolation

Crypt/gland isolation and organoid culture was established and maintained as described previously (*Sato et al., 2011*; *Schlaermann et al., 2016*; *Wallaschek et al., 2019*). Murine colon was opened longitudinally, cut into small pieces and washed with ice-cold chelation buffer (distilled water containing 5.6 mM $Na_2HPO_4$, 8.0 mM $KH_2PO_4$, 96.2 mM NaCl, 1.6 mM KCl, 43.4 mM sucrose, 54.9 mM D-sorbitol, 0.5 mM DL-dithiotreitol (added freshly). Samples were washed until the content of

chelation buffer was clear. For human stomach, overlying mucosa was removed from the submucosa and connective tissue, cut into small pieces and washed with ice-cold chelation buffer until the supernatant was clear.

Tissue pieces were incubated in 2 mM EDTA ice-cold chelation buffer for 5 to 10 min at RT with occasional shaking. Crypts/glands were released from the tissue using a microscopic slide with applied pressure. Tissue pieces and released crypts/glands were placed in cold basal medium (Advanced Dulbecco's modified Eagle (DMEM)/F12 medium, supplemented with 10 mM HEPES and GlutaMAX (Thermo Fisher Scientific). Pieces were allowed to settle and supernatants containing the crypts/glands were transferred into a conical tube and centrifuged at 200 x g for 5 min at 4°C.

## Culture of mouse colon and human gastric organoids

Isolated mouse colon crypts and human gastric glands were embedded in 50 µl of matrigel (Cat# 356231, Corning) in a 24-well plate. For polymerization, the plates were incubated 10 to 30 min at 37°C. Matrigel drops were overlaid with 500 µl of basal medium (DMEM/F12 medium), supplemented with 10 mM HEPES and GlutaMAX) and 2% B27 supplement (Cat# 12587–010, Thermo Fisher Scientific), 100 µg/ml Primocin (Cat# ant-pm-1, InvivoGen), conditioned medium produced in-house: 50% Wnt3A conditioned medium, 10% R-spondin1 (RSP1) conditioned medium, 10% mNoggin conditioned medium, 1.25 mM N-acetyl-L-cysteine (Cat# A9165, Sigma), 10 µM Rho kinase inhibitor Y-27632 (Cat# M1817, Abmole). For the mouse colon crypts, 50 ng/ml animal-free recombinant human epidermal growth factor (EGF) (Cat# AF-100–15, PeproTech) was added additionally. Complete medium for human gastric glands was additionally supplemented with: 2 µM TGF-ß inhibitor A83-01 (TGF-ßi) (Cat# 2939, Tocris Bioscience), 1 nM Gastrin I (Cat# 3006/1, Tocris Bioscience) and 100 ng/ml FGF10 (Cat# 100–26, PeproTech). Rho kinase inhibitor was included for the first 2 days after splitting.

Wnt3A, RSP1 (kind gift from Henner F Farin, Institute for Tumor Biology and Experimental Therapy, Frankfurt) and mNoggin conditioned medium was produced according to the methods described in *Farin et al. (2012)* and *Wallaschek et al. (2019)*.

## Passaging and differentiation of organoids

Mouse colon organoids (MCO)/human gastric organoids (HGO) were split at a ratio of 1:5/1:6 every 7/14 days, respectively. Medium was refreshed every 2 days.

For differentiation, MCO and HGO were grown in complete medium for 2 days or 4 days, respectively. Differentiation was induced by withdrawal of Wnt3A and RSP1 conditioned medium for the next 5/10 days for MCO and HGO, respectively.

## Immunofluorescence staining and image analysis of organoids

Growth medium was discarded from the wells and 500 µl of cold, cell recovery solution (Cat# 354253, Corning) was added per well. The matrigel drop with organoids was transferred into a 15 ml conical tube and placed on ice for 30–45 min or until matrigel dissolves. Cold DMEM/F12 medium was added and samples were centrifuged at 100 x g for 3 min at 4°C. Organoids were fixed in 4% PFA for 20 min at RT. Next, organoids were blocked and permeabilized with 500 µl of DPBS/T/D (1x DPBS (Cat# 14190–094, Gibco), containing 0.3% Triton X-100 (AppliChem), 1% DMSO (AppliChem) and 1% BSA (Sigma) for 1 hr at RT. Organoids were allowed to sink in by gravity, supernatants were discarded and organoids were incubated overnight at 4°C with primary antibody against Keratin 20 (Krt20) (dilution 1:250) in 5% normal goat serum (Cat# 31873, Thermo Fisher Scientific) and DPBS/T/D. Next day, organoids were washed 4 x with 1% BSA in DPBS and incubated for 3 hr at RT with anti-mouse IgG-Cyanine3 (Cat# A10521, Thermo Fisher) at 8 µg/ml in 5% normal goat serum and DPBS/T/D), Hoechst staining was added at 10 µg/ml final concentration for 10 min. Organoids were washed 4 x in 1% BSA/DPBS.

Samples were analyzed with confocal microscopy using Leica SP-8 microscope (20x, dry objective) and image processing was done using Leica confocal software. Images are presented with maximum projection.

## CRISPR/Cas9-mediated ERK3 knockout in human gastric organoids (HGO)

HGO were cultured as described in 'Passaging and differentiation of organoids'. Organoids were cultured without antibiotics for 2 days before infection. Organoids were trypsinized for 10 min at 37°C using TrypLE Express (Cat# 12605028, Thermo Fisher Scientific) to obtain single cells. Single cells were infected with lentiviral particles in the presence of 10 µg/ml of polybrene (Cat# sc-134220, Santa Cruz) and transferred into a 48-well plate, spun down at 600 x g for 1 hr at 32°C followed by 4 hr incubation at 37°C. Next, single cells were collected, resuspended in matrigel and cultivated under normal conditions in complete medium as described in 'Mouse colon and human gastric organoids culture'. After 48 hr, selection was started with 5 µg/ml puromycin (Cat# 0240.3, Carl Roth). Medium was exchanged every other day for 14 days with puromycin. Organoids were split and cells were collected for western blot analysis and knockdown verification.

## Statistical analyses

p-Values were obtained by t-tests (paired test used always for fold change analyses), one-way ANOVA and Turkey's post-test or two-way ANOVA and Bonferroni post-tests in Prism 5.0a and $p < 0.05$ was considered as a significant difference.

## Acknowledgements

We thank Stefanie Wenzel for excellent technical assistance. This work is supported from the grants from Else Kroener Fresenius Stiftung; MERCK (project ID-ERK-KR), CRC 1292, TP12 and Forschungszentrum für Immuntherapie of the University Medical Center Mainz to KR. KR is supported through a Heisenberg professorship of the DFG (RA1739/4-1). We thank the Translational oncology team of MERCK for their valuable inputs and advise. We thank Dr. Christiane Schoenfeld for help with the animal experiments and for critical reading of the mansucript. Hermann Goetz was thanked for the quantification of lung nodules. The work of FM is supported by the German Federal Ministry of Education and the University Medical Center Mainz; MPR is funded by „Forschungszentrum Immuntherapie", University of Mainz („Naturwiss. Medizinisches Forschungszentrum "to MPR), Stiftung „Lichterzellen "to MPR and Research (BMBF 01EO1003 Deutsche Forschungsgemeinschaft (TRR156/1 to MPR).

## Additional information

### Funding

| Funder | Grant reference number | Author |
|---|---|---|
| Deutsche Forschungsgemeinschaft | RA1739/4-1 | Krishnaraj Rajalingam |
| Deutsche Forschungsgemeinschaft | RA1739/4-2 | Krishnaraj Rajalingam |
| Deutsche Forschungsgemeinschaft | CRC1292 | Katarzyna Bogucka |
| Merck KGaA | ERK-KR | Krishnaraj Rajalingam |
| Else Kröner-Fresenius-Stiftung | SUNMAPK | Malvika Pompaiah |
| German Federal Ministry of Education | | Federico Marini |
| University Medical Center Mainz | | Federico Marini |
| Else Kröner-Fresenius-Stiftung | SUNMAPK | Katarzyna Bogucka |

The funders had no role in study design, data collection and interpretation, or the decision to submit the work for publication.

## Author contributions
Katarzyna Bogucka, Conceptualization, Data curation, Formal analysis, Validation, Investigation, Methodology, wiriting-original draft, reviewing-editing; Malvika Pompaiah, Formal analysis, Investigation, Methodology; Federico Marini, Data curation, Formal analysis; Harald Binder, Formal analysis, Supervision; Gregory Harms, Software, Formal analysis, Methodology; Manuel Kaulich, Peter Grimminger, Resources, Methodology; Matthias Klein, Christian Michel, Methodology; Markus P Radsak, Supervision, Methodology; Sebastian Rosigkeit, Investigation, Methodology; Hansjörg Schild, Conceptualization, Supervision; Krishnaraj Rajalingam, Conceptualization, Supervision, Funding acquisition, Writing - original draft, Project administration, Writing - review and editing

## Author ORCIDs
Federico Marini ⓘ http://orcid.org/0000-0003-3252-7758
Manuel Kaulich ⓘ http://orcid.org/0000-0002-9528-8822
Krishnaraj Rajalingam ⓘ https://orcid.org/0000-0002-4175-9633

## Ethics
Human subjects: Tissue samples employed are obtained from the biobank of the university medical center. Written informed consent was obtained from all patients, and the study was approved by the ethical committee at the University Medical Center of the JGU Mainz (approval # 837.100.16 (10419)).
Animal experimentation: The animal experiment was performed under the permission (G16-1-026 and AZ23 177-07/G17-1-036) of the National Investigation Office Rheinland-Pfalz and conducted according to the German Animal Protection Law.

## Decision letter and Author response
Decision letter https://doi.org/10.7554/eLife.52511.sa1
Author response https://doi.org/10.7554/eLife.52511.sa2

# Additional files

## Supplementary files
• Transparent reporting form

## Data availability
The RNA-seq data presented in this manuscript have been deposited in NCBI's Gene Expression Omnibus and are accessible through GEO series accession number GSE136002 (https://www.ncbi.nlm.nih.gov/geo/query/acc.cgi?acc=GSE136002).

The following dataset was generated:

| Author(s) | Year | Dataset title | Dataset URL | Database and Identifier |
|---|---|---|---|---|
| Bogucka | 2020 | control vs siERK3 RNA seq analysis | https://www.ncbi.nlm.nih.gov/geo/query/acc.cgi?acc=GSE136002 | NCBI Gene Expression Omnibus, GSE136002 |

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
