## [Decision Letter]

[Editors’ note: the authors submitted for reconsideration following the decision after peer review. What follows is the decision letter after the first round of review.]

Thank you for submitting your work entitled "ERK3/MAPK6 controls IL-8 production and chemotaxis" for consideration by *eLife*. Your article has been reviewed by two peer reviewers, one of whom is a member of our Board of Reviewing Editors, and the evaluation has been overseen by a Senior Editor. The reviewers have opted to remain anonymous.

Our decision has been reached after consultation between the reviewers. Based on these discussions and the individual reviews below, we regret to inform you that your work will not be considered further for publication in *eLife* at this stage.

The authors have carried out experiments to show the existence of an ERK3-AP-1/c-Jun axis in modulating IL-8 secretion, which plays a role in neutrophil chemotaxis. The assays and reagents chosen to explore the molecular mechanisms were reasonable and reliable. However, the reviewers felt that unless the contribution of this pathway is shown to be associated with some patho-physiological process (e.g. murine tumor metastasis or microbial infection model), the impact of the work is more limited. It remains unknown whether defects in the neutrophil recruitment are directly linked to insufficient IL-8 production (could be other cytokines/chemokines). To nail down the conclusion and importance of this study, KO mice or local injection of viral vector based shRNA will be necessary.

The reviewers were concerned that it would be challenging to complete these experiments in two months’ time, as is stipulated by *eLife*. In addition, most of the data were generated with a few cell lines in ex vivo settings. Uncertainty remains whether the conclusions can be generalized (e.g. freshly isolated primary colon cancer cells versus adjacent normal epithelial cells would be more convincing). We would of course be happy to reconsider a new manuscript that includes such experiments that were viewed as potentially increasing the overall impact significantly.

Reviewer #1:

The authors spent efforts to show the existence of ERK3-AP-1/c-Jun axis in modulating IL-8 secretion in epithelial cells, which can play a role in neutrophil chemotaxis.

1) It remains largely unknown whether this signaling participates in or contributes to any patho-physiological processes.

2) The authors need to provide more experimental evidence on the importance of this signaling pathway. It is not clear whether other ERK members also contributes to LPS-induced IL-8 secretion, knockdown or inhibition strategies are necessary. Is AP-1/c-Jun the only downstream route of ERK3 to promote LPS-triggered IL-8 secretion? It is also not clear how ERK3 activate AP-1/c-Jun, though its kinase activity is not required in this process?

3) Most of the data were generated with few cell lines in ex vivo settings. Have the authors considered testing freshly isolated primary cells (colon cancer cells versus adjacent normal epithelial cells)? In addition, the differences in ERK3 expression/activation/stability between tumor cells and epithelial cells upon LPS stimulation should be explored more carefully. Can other TLR agonists or PRR ligands induce similar changes? Do they share the same signaling cascade?

4) ERK3 deficient cells showed defects to recruit monocytes and neutrophils, and this is due to insufficient chemokine/cytokine production. Have they considered to check the impact of these changes on tumor progression/metastasis, or infection related settings?

Reviewer #2:

The authors of the manuscript "ERK3/MAPK6 controls IL-8 production and chemotaxis" investigate the regulatory role of ERK3. They show that ERK3 abundance in response to LPS is differentially regulated in tumorigenic and primary cells. They show that ERK3 is critical for the c-Jun/AP-1 dependent expression of IL-8, important for chemotaxis of neutrophils and monocytes and finally touch upon a crosstalk between the canonical MAPKs and ERK3 in IL-8 regulation. The manuscript is well written and methods are technically sound.

The role of TLR4 in the LPS-mediated regulation of ERK3 should be addressed experimentally in order to delineate the pathway from the receptor to the effector.

Interactome studies of ERK3 in the presence and absence of LPS activation might yield further insights into physical protein-protein interactions and possible crosstalks with other MAPK pathways.

Some minor textual editing is necessary, to correct some typos/duplications like for example in the second paragraph of the Introduction.

As a good scientific practice representative immunoblots should always be accompanied by their original full scans (in the supplement or online data).

---

## [Author Response]

[Editors’ note: The authors appealed the original decision. What follows is the authors’ response to the first round of review.]

Reviewer #1:The authors spent efforts to show the existence of ERK3-AP-1/c-Jun axis in modulating IL-8 secretion in epithelial cells, which can play a role in neutrophil chemotaxis.1) It remains largely unknown whether this signaling participates in or contributes to any patho-physiological processes.

We believe that our discovery of ERK3-IL-8 signaling axis in regulating epithelial function and chemotaxis is the first step into the better understanding how ERK3 is regulated and its contribution towards progression of both neoplastic and immune diseases. We appreciate the concern of the reviewer to validate the relevance of these observations in patho/physiological conditions. We have now added substantial evidence to support our main claims:

1) By employing mouse colon organoids (MCOs) and human gastric organoids (HGOs) we demonstrate that ERK3 protein levels are reduced under differentiated conditions. In addition, CRISPR-mediated knockout of ERK3 in HGOs failed to prevent the 3D growth of human gastric epithelium (new Figure 1), further indicating that ERK3 is a potential druggable target. This is also in line with the new mouse knockout data (Ronkina et al., 2019; Soulez et al., 2019). In addition, we further present evidence that loss of ERK3 led to a reduction in IL-8 secretion in HGOs (new Figure 4L).

2) ERK3 regulates IL-8 production in many tested cell lines, including MDA-MB231 (Figure 4G-H and Figure 5A). IL-8 has been reported to enhance migration and thus metastasis of breast carcinoma cells MDA-MB231 (Jayatilaka et al., 2017). To further validate our observations we tested for the lung colonization of these cells upon ERK3 depletion. Consistent with our observations, ERK3-depleted cells exhibited reduced lung metastasis. These data are now added to the manuscript (new Figure 5C-F and Figure 5—figure supplement 1).

3) We would like to remind that we have also demonstrated IL-8 dependent in vivo chemotaxis of granulocytes and monocytes towards ERK3-depleted culture supernatants of primary epithelial cells by employing a mouse model (Figure 8)

2) The authors need to provide more experimental evidence on the importance of this signaling pathway. It is not clear whether other ERK members also contributes to LPS-induced IL-8 secretion, knockdown or inhibition strategies are necessary.

We have indeed performed extensive analyses on the cross-talk between the classical MAPKs and ERK3, including ERK1/2 and it is included in the manuscript (“Cross-talk between canonical MAPKs and ERK3 in the regulation of IL-8” is a section in our manuscript). Please see Figure 4—figure supplement 3.

To summarize our data:

To explore the role of canonical MAPKs in the regulation of epithelial function we checked for the potential cross-talk between ERK1/2 and ERK3 in controlling IL-8 levels as several studies unveiled a critical role for classical MAPK in the production and secretion of IL-8 (Hartman et al., 2017; Lee et al., 2006; Marie et al., 1999). We performed experiments with trametinib (a well-established MEK1/2 inhibitor) to evaluate the role of canonical MAPKs-ERK1/2 and showed that blocking of the MEK1/2-ERK1/2 pathway leads to a decrease in IL-8 production (Figure 4—figure supplement 3A-B). Moreover, in the context of ERK3, our data clearly showed that inhibition of the ERK1/2 activity leads to a significant downregulation of ERK3 protein but not the mRNA levels (Figure 4—figure supplement 3A-C). We further demonstrated that treatment with MG132 prevented ERK3 degradation and rescued IL-8 levels inhibited by the trametinib treatment (Figure 4—figure supplement 3D and E).

We also present evidence with other innate immune stimuli like IL-1β, R848 and PamCSK3 and we found cell specific and stimulus specific effects with respect to the requirement of ERK3 in stimulus-dependent IL-8 production. These studies deserve follow up to delineate the role of various MAPKs in mediating epithelial chemotaxis and their potential cross-talk with ERK3.

Is AP-1/c-Jun the only downstream route of ERK3 to promote LPS-triggered IL-8 secretion?

As presented in Figure 6—figure supplement 1 and Figure 6—source data 1, ERK3 is required for the activation of majority of the tested TFs in HCPECs, including AP-1 (Figure 6D), which has been reported to control both, basal and inducible expression of chemokines like IL-8 (Khanjani et al., 2012; O'Hara et al., 2009). Hence, AP-1/c-Jun became the major focus in this study. Nevertheless, we independently validated TF array-derived data by assessing the activity of three most potent regulators of IL-8 (Hoffmann et al., 2002; Jundi and Greene, 2015) in ERK3-depleted cells, which apart from AP1/c-Jun, also involves C/EBP and CREB. Our results highlighted AP-1 as one the TFs positively regulated by ERK3 and therefore regulating IL-8 transcription (new Figure 6—figure supplement 2).

We cannot with certainty state that ERK3 regulates IL-8 production exclusively through AP-1/c-Jun. Considering that the production of IL-8 is regulated at several levels, including transcriptional regulation, we focused on verifying the role of ERK3 in the transcriptional regulation of IL-8, which unveiled the previously unknown ERK3-AP-1/c-Jun-IL-8 signaling axis.

It is also not clear how ERK3 activate AP-1/c-Jun, though its kinase activity is not required in this process?

We demonstrated that ERK3 is critical for AP-1 signaling through its interaction and regulation of c-Jun protein and its nuclear abundace (Figure 7).

In order to activate target genes, transcription factors such as c-Jun have to translocate to the nucleus (Schreck et al., 2011). Our studies revealed that ERK3 is required for nuclear abundance of c-Jun protein (Figure 7B and E-F). We provide evidence that ERK3 interacts with c-Jun and regulates the DNA binding activity of AP-1 (Figure 6D-E, G and Figure 7) which might reflect a non-catalytic scaffolding function described for other kinases as well (Rauch et al., 2011). Sub-nuclear localization of AP-1 proteins is crucial for the AP-1 activity (Gazon et al., 2017), therefore it is tempting to propose ERK3 as a novel MAPK required for the nuclear abundance of c-Jun.

3) Most of the data were generated with few cell lines in *ex vivo* settings. Have the authors considered testing freshly isolated primary cells (colon cancer cells versus adjacent normal epithelial cells)?

Well taken! We employed commercially available Human Primary Epithelial Cells (HCPECs) as the primary colonic epithelial cells and performed majority of the experiments in these cells apart from the HT-29 colon adenocarcinoma cells. Obtaining duly consented patient tissue for such experiments takes months at least in our clinical center. We therefore approached commercial biobank to obtain such normal and colon carcinoma tissue with clean pathology from duly consented patients. We were able to obtain EpCAM sorted epithelial-like cells from tumor and adjacent normal tissue. We attempted to culture these cells and perform ERK3 knockdown studies as well as LPS stimulation. Unfortunately, these cells failed to attach to various substrates and we still pursue to get a clone out of these cells to perform such experiments. In the meantime, we directly lysed some of the sorted normal vs. tumor cells to verify ERK3 protein expression. As you can see in Author response image 1, EpCAM positive cells isolated from normal tissue expressed higher levels of ERK3 protein than the matching tumor epithelial cells.

**Author response image 1. respfig1:** Western Blot analyses of colon epithelial-like cells isolated using EpCAM microbeads from tumor and adjacent normal tissue of two CRC patients. Cells were analysed by Western blot according to the Materials and methods section, expression levels of ERK3 were monitored.

These data are very preliminary as they represent only two patients and alone it is insufficient to draw any general conclusions. Our unpublished observations reveal that this phenotype is also reflected in the mRNA levels i.e. ERK3 levels are downregulated in colorectal cancer (CRC) samples in comparison to the matching normal or adjacent normal controls. Whether this has any functional significance in the context of CRC- immune-tumour micromilieu needs a complete study on its own.

In the meantime, we were able to establish mouse and human gastrointestinal organoids and we did confirm that loss of ERK3 led to a reduction in the secretion of IL-8 in HGOs derived from two patients (new Figure 1 and new Figure 4L).

In addition, the differences in ERK3 expression/activation/stability between tumor cells and epithelial cells upon LPS stimulation should be explored more carefully. Can other TLR agonists or PRR ligands induce similar changes? Do they share the same signaling cascade?

The data involving other innate immune stimuli in HCPECs have been included already in the manuscript. However, to further strengthen these studies we employed TLR1/2 ligand-Pam3CSK4, TLR7/9 ligand-R848 and one of the most potent innate immune stimuli-IL-1β and monitored ERK3 kinetics in both HCPECs and HT-29 cells (new Figure 2—figure supplement 3).

Additionally, we tested the effect of IL-1β and TLR1/2 ligand Pam3CSK4, two potential ERK3 stimuli in the production of IL-8 by depleting ERK3 from HCPECs and HT-29 by si- shRNA (new Figure 4—figure supplement 2). IL-1β induced IL-8 levels in both cell types (new Figure 4—figure supplement 2A-B and E-F), however, knockdown of ERK3 negatively affected observed upregulation in chemokine production only in HT-29 cells (new Figure 4—figure supplement 2E-F). Conversely, Pam3CSK4 induced IL-8 levels in HCPECs and depletion of ERK3 prevented this upregulation (new Figure 4—figure supplement 2C-D), while almost no effect of TLR1/2 ligand was detected in HT-29 cells (new Figure 4—figure supplement 2G-H). These data confirmed again the cell type and stimuli specific activation of ERK3 and its requirement for IL-8 production.

4) ERK3 deficient cells showed defects to recruit monocytes and neutrophils, and this is due to insufficient chemokine/cytokine production. Have they considered to check the impact of these changes on tumor progression/metastasis, or infection related settings?

As mentioned in response to the comment #1: Considering that ERK3 regulates IL-8 production in many tested cell lines, including MDA-MB231 (Figure 4G-H and Figure 5A), we investigated the metastatic potential of control (shCo) and ERK3-depleted (shERK3) MDA-MB231 cells (new Figure 5C-F). Intravenous (i.v.) injection of ERK3-depleted (shERK3) MDA-MB231 cells resulted in less tumor lesions in the lungs and decreased pulmonary metastatic burden (new Figure 5E and F, respectively and Figure 5—figure supplement 1, Results subsection “Depletion of ERK3 reduces metastatic potential of breast cancer cell line MDA-MB231”).

Reviewer #2:[…] The role of TLR4 in the LPS-mediated regulation of ERK3 should be addressed experimentally in order to delineate the pathway from the receptor to the effector.

We thank the reviewer for his/her support.We appreciate the comment of the reviewer that the entire pathway from the receptor to the effector needs to be established which is a manuscript in itself. We have now performed extensive analysis to strengthen our main arguments confirming the physiological and patho-physiological relevance of ERK3-IL-8 signaling axis as also advised in the main decision letter. We have also extended our studies to other ligands (R848, PamCSK4 and IL-1β) in response to the request from reviewer 1 (new Figure 2—figure supplement 3 and Figure 4—figure supplement 2).

We also demonstrate high expression of TLR4 in tumorigenic cells in comparison to human primary cells (Figure 2—figure supplement 2), which might influence the sensitivity to ligands like LPS. Observed discrepancies in responsiveness to LPS between colon carcinoma and primary epithelial cells has been already reported and we referred to these publications in our Discussion.

We indeed tested that loss of TLR4 prevented ERK3 activation in HT-29 cells and thus IL-8 secretion, confirming the specificity of the response (these data are not shown). If needed, we can share it with the reviewers.

Interactome studies of ERK3 in the presence and absence of LPS activation might yield further insights into physical protein-protein interactions and possible crosstalks with other MAPK pathways.

Good suggestion indeed! The interactome studies and subsequent validation to delineate the role of these partners in activation of TLR signaling with respect to MAPKs deserves further studies and we hope that publishing these first observations unveiling a role for atypical MAPK ERK3 in the control of epithelial secretome opens new lines of investigations as proposed. This is clearly the next stage of the project.

Also to identify the E3 ligases, the kind of Ubiquitin chains and the site of ubiquitination needs to be followed up. The cell type and stimuli specific activation of ERK3 and the functional significance are exciting questions for the future. We hope that this work will also open further lines of investigations along these lines.

Please note that cross-talk between the classical MAPK and ERK3 has been included in the manuscript. Please see Figure 4—figure supplement 3, subsection “Cross-talk between canonical MAPKs and ERK3 in the regulation of IL-8”.

We hope that the informed reviewers allow the publication of these already data dense manuscript unveiling the role for ERK3 in the control of AP-1-IL-8 signaling axis contributing to epithelial chemotaxis and metastasis.